# Desmoplastic stroma restricts T cell extravasation and mediates immune exclusion and immunosuppression in solid tumors

Zebin Xiao [1], Leslie Todd [1], Li Huang[1], Estela Noguera-Ortega [2], Zhen Lu[1], Lili Huang[3], Meghan Kopp[1], Yue Li[1], Nimisha Pattada[1], Wenqun Zhong[4], Wei Guo [4], John Scholler[5], Maria Liousia[2], Charles-Antoine Assenmacher[6], Carl H. June [5], Steven M. Albelda[2] & Ellen Puré [1] ✉

The desmoplastic stroma in solid tumors presents a formidable challenge to immunotherapies that rely on endogenous or adoptively transferred T cells, however, the mechanisms are poorly understood. To define mechanisms involved, here we treat established desmoplastic pancreatic tumors with CAR T cells directed to fibroblast activation protein (FAP), an enzyme highly overexpressed on a subset of cancer-associated fibroblasts (CAFs). Depletion of FAP[+] CAFs results in loss of the structural integrity of desmoplastic matrix. This renders these highly treatment-resistant cancers susceptible to subsequent treatment with a tumor antigen (mesothelin)-targeted CAR T cells and to anti-PD-1 antibody therapy. Mechanisms include overcoming stroma-dependent restriction of T cell extravasation and/or perivascular invasion, reversing immune exclusion, relieving T cell suppression, and altering the immune landscape by reducing myeloid cell accumulation and increasing endogenous CD8[+] T cell and NK cell infiltration. These data provide strong rationale for combining tumor stroma- and malignant cell-targeted therapies to be tested in clinical trials.

Immunotherapies that rely on endogenous or adoptively transferred T cells have proven to be challenging in treating solid tumors[1–3]. To date, attempts to improve the efficacy of chimeric antigen receptor (CAR) T cells in solid tumors have focused on modifications to improve their intrinsic functionality, to increase their resistance to immunosuppressive factors, and to enhance trafficking into tumors[4]. Unfortunately, attempts at translating these strategies to the clinic have so far proved to be disappointing[5]. One reason for poor efficacy of tumor-

associated antigen (TAA)-targeted immunotherapies in solid tumors is the complex tumor microenvironment (TME) that insulates tumor cells against effective cytotoxic immune responses[4]. Understanding and overcoming the barriers that exclude T cell infiltration and/or inhibit T cell function will be critical to achieving sustained efficacy of immunotherapies in most solid tumor patients.

Key components of the immunosuppressive TME in solid tumors include cancer-associated fibroblasts (CAFs) and the associated

[1]Department of Biomedical Sciences, University of Pennsylvania, Philadelphia, PA 19104, USA. [2]Department of Medicine, Perelman School of Medicine, University of Pennsylvania, Philadelphia, PA 19104, USA. [3]Department of Pathology and Laboratory Medicine, Perelman School of Medicine, University of Pennsylvania, Philadelphia, PA 19104, USA. [4]Department of Biology, School of Arts & Sciences, University of Pennsylvania, Philadelphia, PA 19104, USA. [5]Center for Cellular Immunotherapies, Perelman School of Medicine, University of Pennsylvania, Philadelphia, PA 19104, USA. [6]Department of Pathobiology, School of Veterinary Medicine, University of Pennsylvania, Philadelphia, PA, USA. ✉e-mail: epure@upenn.edu

desmoplastic matrix[6,7]. Previous studies demonstrated that depletion or reprogramming of certain subsets of stromal cells in solid tumors can disrupt the pro-tumorigenic niche and enhance endogenous and vaccine-induced anti-tumor immunity[8–13]. One subset of CAFs that appears to be important in this process expresses the surface protease fibroblast activation protein (FAP)[14,15]. Indeed, consistent with the hypothesis that this subset of stromal cells makes a significant contribution to the physical barriers and functional inhibition of anti-tumor immunity, we and others demonstrated that CAR T cells targeted to fibroblast activation protein (FAP-CAR T) on FAP-expressing stromal cells can effectively inhibit tumor growth in mouse models of solid tumors[9,11,16]. Depletion of FAP+ stromal cells was shown to inhibit tumor growth via immune-dependent and immune-independent mechanisms in a context-dependent manner related to tumor immunogenicity and the extent of desmoplasia[9], however, the mechanisms involved have not been defined. Moreover, in a recent study of lung cancer, a subpopulation of FAP+ CAFs localized within the stroma that form multiple layers around tumor nests, were posited to play a role in T cell exclusion from tumor nests through deposition and alignment of matrix fibers[17]. The authors further posited that targeting of this subset of CAFs should enhance the efficacy of immunotherapy in the context of T cell-excluded tumors.

The role of tumor stroma as a critical component of the TME that contributes to tumor growth and resistance to therapy is particularly striking in pancreatic ductal adenocarcinoma (PDAC), in which stroma can represent >90% of the tumor volume and outcomes of any therapy remain dismal[13,18–22]. The presence of distinct tumor nests has allowed the classification of the immune status of PDAC tumors toward a paucity of T cell infiltration (referred to as immune deserts) or accumulation of T cells in stroma-rich peri-tumoral regions but, excluded from tumor nests (referred to as immune exclusion). These peri-tumoral T cells are subject to immune suppression[22,23] and this tumor exclusion phenomenon appears to contribute to the limited efficacy of adoptively transferred TAA-targeted CAR T cells[22,24,25].

In this study, we show three mechanisms that contribute to the efficacy of FAP-CAR T cells, compared to TAA-targeted mesothelin (Meso)-CAR T cells, in inhibiting tumor growth. First is their capacity to eliminate FAP+ stromal cells, thereby overcoming restriction of their extravasation and penetration of perivascular regions. Second is their ability to deplete the desmoplastic peri-tumoral stroma resulting in reversal of immune exclusion, and finally, their capacity to reprogram the immunosuppressive milieu, thus, rendering the TME permissive to TAA-directed endogenous T cell and adoptively transferred CAR T cell tumoricidal activity. In comparison, TAA-targeted Meso-CAR T cells are largely excluded from tumors and the few tumor-infiltrating Meso-CAR T cells detected are functionally suppressed in the context of intact stroma. Importantly, dual sequential treatment with FAP-CAR T cells overcomes the negative effects of the TME on Meso-CAR T cells or immune checkpoint blockade in pre-clinical PDAC models, resulting in significant inhibition of tumor growth in multiple PDAC models by anti-PD-1 and by Meso-CAR T cells. This study establishes a strategy for potentiating immunotherapy in a wide variety of desmoplastic tumors based on initial disruption of the tumor stroma followed by TAA-CAR T cells or immune checkpoint blockade therapy (ICT) in settings in which they are poorly, if at all, effective as monotherapies.

## Results

### Comparable transduction efficiency, CAR expression and function of FAP- and Meso-CAR T cells in vitro

FAP is highly expressed in stroma of most carcinomas, including PDAC[9], and mesothelin is a promising TAA target expressed at high levels in PDAC (Supplementary Fig. 1a)[24,26]. To compare the behavior of tumor stroma-targeted FAP-CAR T cells with TAA (mesothelin)-targeted CAR T cells, we utilized a second-generation FAP-CAR vector encoding a scFv based on the sequence of the anti-FAP monoclonal

antibody 4G5 produced in our laboratory that cross reacts with mouse, human and canine FAP[27], and a previously described mouse Meso-CAR designated A03[24] (Fig. 1a). Empty retroviral MigR vector-transduced T cells were used as a negative control[9,11]. All 3 vectors encoded an IRES sequence followed by an enhanced green fluorescent protein (EGFP) cassette. The FAP and Meso CARs encoded 4G5 and A03 scFvs respectively, followed by a CD8 transmembrane/hinge sequence and 4-1BB and CD3ζ signaling domains (Fig. 1a). Mouse splenocytes were isolated, activated, and transduced with MigR control, Meso-CAR and FAP-CAR, consistently resulting in greater than 90% transduction efficiencies (Fig. 1b). Moreover, transduced MigR control, Meso-CAR and FAP-CAR T cells exhibited similar levels of proliferation following activation with anti-CD3/CD28 beads (Supplementary Fig. 1b). Flow cytometry analyses of the GFP+ transduced T cells showed granzyme B (GzmB), tumor necrosis factor-α (TNF-α), and interferon-γ (IFN-γ) were also upregulated to a similar extent in the GFP+ T cells transduced with MigR, Meso-CAR and FAP-CAR (Supplementary Fig. 1c). Finally, the chemokine, CXCL9, induced comparable transmigration of activated MigR control, Meso-CAR, and FAP-CAR T cells in a dual chamber migration assay (Supplementary Fig. 1d).

To compare the effector function of Meso- and FAP-CAR T cells in vitro, we assessed their cytolytic activities and IFN-γ release following co-culture with cells expressing their respective antigen targets, relative to their respective negative control targets. Specifically, Meso-CAR T cells were co-cultured with mesothelin-expressing parental 4662 PDAC cells or with mesothelin-knockout (KO) 4662 PDAC cells generated using *CRISPR/Cas9* system (Supplementary Fig. 1a), while FAP-CAR T cells were co-cultured with FAP-negative 3T3 fibroblasts or with 3T3 fibroblasts transfected to express mouse FAP[11]. Meso-CAR T and FAP-CAR T cells exhibited comparable antigen- and dose-dependent cytotoxic activity (Fig. 1c) and activation, as measured by IFN-γ release (Fig. 1d) in vitro.

### Stromal-targeted FAP-CAR T cells inhibited PDAC tumor growth more effectively than Meso-CAR T cells in vivo

To compare the anti-tumor activity of FAP- and Meso-CAR T cells in vivo, we established subcutaneous 4662 PDAC tumors in syngeneic C57BL/6 mice. Mesothelin (Supplementary Fig. 1a) and FAP are highly expressed in 4662 tumor cells and tumor stromal cells, respectively[9]. To evaluate anti-tumor efficacy, we administered $5 \times 10^6$ FAP-CAR+, Meso-CAR+ and a comparable number of total MigR control T cells or PBS intravenously in mice when tumors reached a mean volume of 100–150 mm³ (Fig. 1e). Compared with PBS or MigR control T cells, Meso-CAR T cells only minimally inhibited tumor growth of tumor-bearing mice. More robust inhibition of tumor growth and more prolonged survival was observed in tumor-bearing mice treated with FAP-CAR T cells. Importantly, treatment did not result in evident toxicity, including no significant change in body weight (Fig. 1f–i and Supplementary Fig. 2a–c).

Similar results were obtained in studies using the KPC genetically engineered mouse model (GEMM) that recapitulates tumor initiation, progression, and the genetic and histopathological characteristics of human PDAC[28]. KPC mice bearing spontaneous tumors of similar size, based on longitudinal non-invasive ultrasound imaging, were treated with $5 \times 10^6$ MigR control, Meso-CAR, and FAP-CAR T cells (Fig. 1j). In this model, we again saw only modest effects from the Meso-CAR T cells but observed more robust inhibition of tumor growth with the FAP-CAR T cells compared with MigR control and Meso-CAR T cells, monitored by non-invasive ultrasound imaging (Fig. 1k and Supplementary Fig. 2d–g). Median and overall survival were extended by both Meso-CAR T and FAP-CAR T cells with a more profound effect observed in the FAP-CAR T cell treated cohort (Fig. 1l).

To further address the potential for on-target, off-tumor effects of 4G5 FAP-CAR T cells, we treated 4662 PDAC tumor-bearing syngeneic C57BL/6 mice with two doses of MigR control or FAP-CAR T cells and

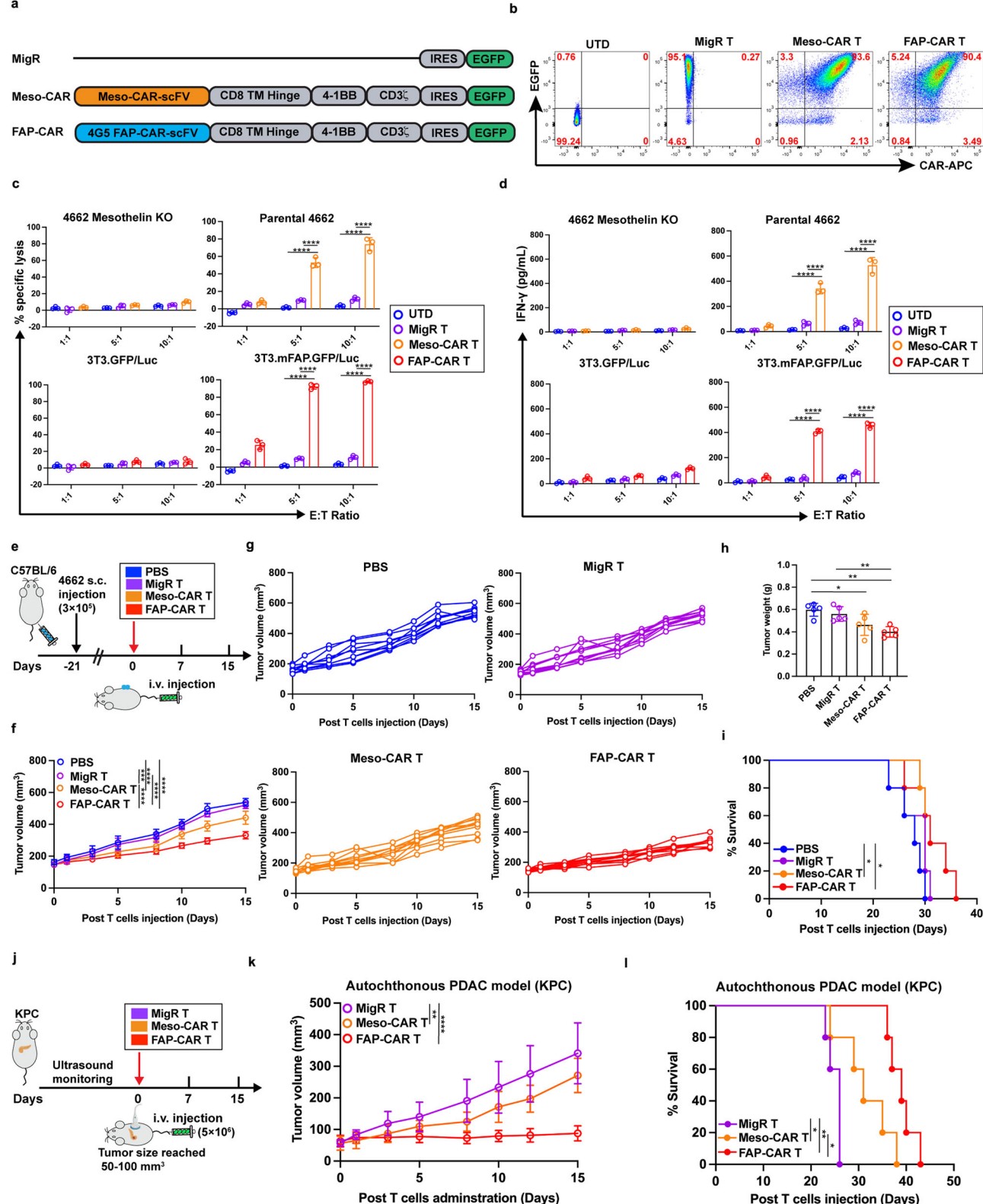

collected multiple organs from one cohort at 1 week after the first dose and from a second cohort at 1 week after the second dose. Organ pathology compared to non-tumor-bearing untreated controls was assessed blindly by a certified veterinary pathologist (C-A.A.) (Supplementary Fig. 3a). We found minimal bone marrow hypoplasia in 2 out of 5 mice that received a single dose, and no bone marrow changes in any of the mice one week after two doses of FAP-CAR T cells. Mild to moderate splenic white pulp hyperplasia was observed in 60% of mice

that received a single dose and 100% of mice given two doses of FAP-CAR T cells. Mild to moderate mononuclear infiltrate of quadricep muscle, pancreas (with a single mouse exhibiting more severe infiltrate) and lung were observed at 1 week post a single injection of FAP-CAR T cells, but these effects appeared to be transient as they, for the most part, resolved by 1 week post a second dose of FAP-CAR T cells (Supplementary Fig. 3b). Although changes in ALP, RBC, HGB, and HTC were noted, they were similar between control MigR and FAP-CAR T

**Fig. 1 | Stromal-targeted FAP-CAR T cells inhibit PDAC tumor growth more effectively than Meso-CAR T cells. a** Depiction of control MigR, Meso-CAR and FAP-CAR recombinant MigR constructs. **b–d** Comparable transduction efficiency, CAR expression and function of FAP- and Meso-CAR T cells in vitro**. b** Flow cytometric analysis of EGFP expression (Y-axis) and reactivity with CAR binding APC-conjugated anti-F(ab')$_2$ fragment antibodies (X-axis). **c** Cytotoxic activity and **d** IFN-γ release when T cells were cultured with target cells expressing their specific targets, Meso$^+$ 4662 tumor cells and murine FAP transfected 3T3 cells, relative to negative control Meso-KO 4662 cells and FAP-negative parental 3T3 cells, respectively (*n* = 3 independent samples/group). Data points are mean±SD and groups were compared using two-way ANOVA with Tukey's multiple comparisons test. ****$p$ < 0.0001. **e–l** Impact of FAP-CAR T vs. Meso-CAR T on growth of established 4662 tumors subcutaneously transplanted into syngeneic mice and (**j–l**) established spontaneous pancreatic tumors in KPC mice. **e** Experimental timeline for treatment of 4662 syngeneic PDAC tumor bearing C57BL/6 mice. Average (**f**) and individual (**g**) growth curves, tumor weight at endpoint (**h**) and modified Kaplan-Meier curves (**i**) for each treatment cohort in the syngeneic transplant model. **j** Schematic of KPC autochthonous tumor model experimental design. Average tumor growth curves measured by ultrasound (**k**) and modified Kaplan-Meier plots (**l**) for KPC mice with indicated treatments. Data indicate mean±SD (*n* = 5 per group) and were determined by one-way ANOVA with Tukey's multiple comparisons test (**f**, **h**, and **k**) or log-rank test (**i** and **l**). **j** The pancreas and ultrasound are created with BioRender.com. *$p$ < 0.05, **$p$ < 0.01, ***$p$ < 0.001, and ****$p$ < 0.0001. **h** PBS vs. Meso-CAR, $p$ = 0.032; PBS vs. FAP-CAR, $p$ = 0.002; MigR vs. FAP-CAR, $p$ = 0.010. i. PBS vs. Meso-CAR, $p$ = 0.049; MigR vs. FAP-CAR, $p$ = 0.004; Meso-CAR vs. FAP-CAR, $p$ = 0.020. **k**. MigR vs. Meso-CAR, $p$ = 0.029. The $p$ values for remaining comparisons are all <0.001 or <0.0001. Source data are provided as a Source Data file.

cell treated mice and therefore related to adoptive cell transfer and not to the activity of FAP-CAR T cells per se, as was the case for the changes in WBC as well (Supplementary Fig. 3c). Collectively, minimal sustained toxicity was observed in the tumor-bearing mice treated with FAP-CAR T cells compared to controls.

## FAP-CAR T cells infiltrate and deplete their target cells more effectively than Meso-CART cells in PDAC tumors in vivo

Since FAP- and Meso-CAR T cells exhibited similar functionality in vitro, we hypothesized that the greater anti-tumor activity of FAP-CAR T cells in vivo was at least partially due to their unique ability to overcome the physical barriers and immunosuppressive milieu imposed by FAP$^+$ stromal cells in the TME. To test our hypothesis, we utilized CD45.1 C57BL/6 mice as T cell donors to produce MigR control, Meso-CAR, and FAP-CAR T cells and adoptively transferred these cells via intravenous injection into CD45.2 C57BL/6 PDAC tumor-bearing hosts. Peripheral blood, spleens, and tumors were harvested to compare the trafficking, intra-tumoral localization and functionality of the adoptively transferred T cells in vivo at 1, 4 and 7 days post-T cell administration (Fig. 2a). Flow cytometry analyses revealed comparable accumulation of MigR control, Meso-CAR and FAP-CAR T cells in peripheral blood and spleens and the donor T cells in blood and spleen in each group expressed high levels of activation and proliferation markers, such as CD69 and Ki-67, and low levels of PD-1 (Supplementary Fig. 4a, b), indicative of good functionality of these T cells. These data suggest that all these T cells behave comparably in circulation.

In contrast, major differences were observed between tumor-infiltrating donor T cells from the different groups. Few CD45.1$^+$GFP$^+$ donor T cells infiltrated tumors in the MigR control and Meso-CAR T cell treated mice. In contrast, we observed significant time-dependent accumulation of CD45.1$^+$GFP$^+$ donor T cells in tumors starting as early as 1-day post-administration of FAP-CAR T cells and the tumor-infiltrating FAP-CAR T cells increased over time at least until 7 days post-administration (Fig. 2b and Supplementary Fig. 5a–c). In addition, on day 7, compared with MigR control and Meso-CAR T cells, tumor-infiltrating FAP-CAR T cells expressed lower levels of PD-1 (Fig. 2c), higher levels of Ki-67 (Fig. 2d), and produced higher levels of IFN-γ (Fig. 2e) after isolation and ex vivo stimulation with a cell activation cocktail of PMA and ionomycin, in the presence of Brefeldin A to enhance intracellular accumulation of cytokines.

Flow cytometry of dissociated cells from tumors demonstrated the increase in FAP-CAR T cell infiltration over time correlated with a progressive reduction in FAP$^+$ cells (Fig. 2f [top and middle panels] and Supplementary Fig. 5d), whereas the reduction in mesothelin$^+$ cells was more modest and not evident until day 7 post-administration of FAP-CAR T cells (Fig. 2f [bottom panel] and Supplementary Fig. 5e), consistent with an indirect inhibition of FAP-CAR T cells on the growth and/or survival of FAP-negative tumor cells compared to their effect on FAP$^+$ stromal cells. In contrast, Meso-CAR T cells had no impact on FAP$^+$ cells at any time-point compared to MigR control T cells and only

modestly reduced the proportion of Mesothelin$^+$ tumor cells and with more delayed kinetics, being evident only at day 7 compared to the more rapid impact of FAP-CAR T cells on the proportion of FAP$^+$ stromal cells at all time-points (Fig. 2f). Moreover, we found FAP-CAR T cells also persisted longer than MigR or Meso-CAR T cells at each time point assessed (Supplementary Fig. 6).

In parallel, we utilized multiplex-immunofluorescence (IF) to map the intra-tumoral localization of CAR T cells (GFP$^+$ cells) in vivo and to determine their spatial relationships with their respective FAP$^+$ stromal cell and EpCAM$^+$ tumor cell targets. In line with our flow data, we found accumulation of only rare MigR control and Meso-CAR T cells at days 1, 4 and 7 (Fig. 2g). In contrast, accumulation of FAP-CAR T cells was evident by day 4 and remarkably robust at day 7. In regions with more tumor-infiltrating FAP-CAR T cells, we found significant decreases in FAP$^+$ stromal cells and EpCAM$^+$ tumor cells (Fig. 2g) consistent with our prior studies demonstrating that FAP-CAR T cell-mediated depletion of stromal cells indirectly causes reduced proliferation and increased death of tumor cells[9].

Spontaneous tumors from KPC mice were also stained and showed an abundance of FAP$^+$ stromal cells surrounding PanCK$^+$ tumor cells (Fig. 2h). Staining done 7 days post-T cell injection showed the tumors to be devoid of GFP$^+$ CAR T cell infiltration in mice treated with MigR control and Meso-CAR T cells (Fig. 2h) which notably had no impact on either FAP$^+$ stromal cell or Mesothelin$^+$ tumor cells even by day 7 post-T cell administration. In contrast, we detected many GFP$^+$ FAP-CAR T cells infiltrating into the tumors, which diminished both FAP$^+$ stromal cells and PanCK$^+$ tumor cells (Fig. 2h), further validating the robust anti-tumor activity of FAP-CAR T cells in this autochthonous PDAC model.

Collectively, these results indicate that FAP-CAR T cells more readily traffic to, infiltrate and survive and/or expand in tumors compared to Meso-CAR T cells and that they more effectively deplete their target cells in syngeneic subcutaneous transplanted and autochthonous PDAC models in vivo. These data provide additional evidence that PDAC tumor progression is dependent on FAP$^+$ stromal cells and that targeting the stroma is sufficient to mediate a robust anti-tumor effect[9,11]. Furthermore, these data directly demonstrate that the differential efficacy in vitro and in vivo of FAP-CAR and Meso-CAR T cells appears to reflect differences in their ability to infiltrate and remain functional in the TME.

## Stromal-targeted FAP-CAR T cells more effectively overcome the physical barrier and immunosuppressive TME compared to Meso-CAR T cells

To better understand the distinct behaviors of stromal and TAA-targeted CAR T cells in vivo, we developed and optimized a precision-cut tumor slice-based real-time two-photon microscopy assay. Briefly, mesothelin$^+$ 4662 PDAC cells were transduced with mCerulean3 (Supplementary Fig. 7a)[29] and were subcutaneously inoculated into syngeneic mice and tumor-bearing mice were treated as described

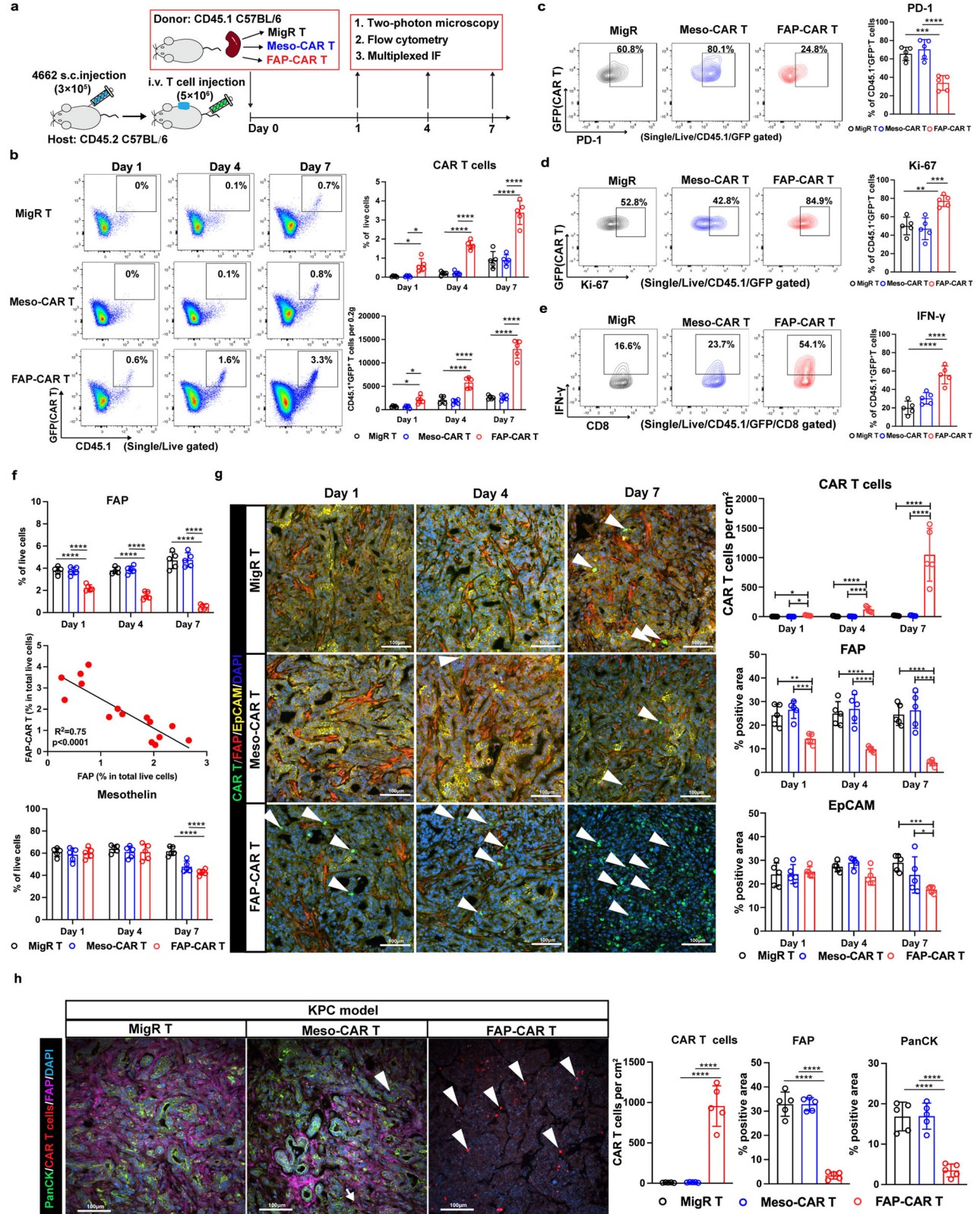

above. Tumors were harvested 1, 4 and 7 days post-administration of EGFP-expressing MigR control, Meso-CAR or FAP-CAR T cells. Tumors were cut into 500 μm slices using a vibratome. The slices were reacted with AF647-conjugated rat anti-mouse CD90.2 to label the stromal cells as described[30] and then immediately analyzed using real-time 2-photon microscopy (Fig. 3a). This ex vivo imaging system overcomes two intrinsic challenges of in situ live-imaging of PDAC tumors:

(1) limited imaging depth resulting from dense stroma, and (2) breathing motion artifacts that limit intravital live imaging[25,31,32]. Importantly, we found that imaging tumor slices from treated mice within 1-2 hours of tumor harvest avoided complications in interpreting data obtained using more standard slice assays in which tumor slices are isolated from untreated mice and then treated in vitro and imaged over several days. Based on preliminary studies, we noted a

**Fig. 2 | FAP-CAR T cells infiltrate and deplete their target cells more effectively than Meso-CAR T cells in PDAC tumors. a** Schematic of experimental protocol. **b** Left: representative flow cytometry plots of tumor-infiltrating MigR control, Meso-CAR, and FAP-CAR T cells. Right: The number of CD45.1⁺GFP⁺ CAR T cells recovered per 0.2 grams of tumor tissue (top) and the percent of CD45.1⁺GFP⁺ CAR T cells amongst total live cells recovered (bottom) at each time point for each group. Representative flow cytometry analysis and quantification of PD-1 **(c)**, Ki-67 **(d)** and IFN-γ **(e)** expression in the tumor-infiltrating MigR control, Meso-CAR and FAP-CAR T cells at day 7 post-administration. **f** Quantification of flow cytometric analysis on FAP⁺ cells (top), Pearson correlation analysis on FAP⁺ cells with the extent of tumor-infiltrating FAP-CAR T cells (middle), and quantification of flow cytometric analysis on mesothelin⁺ cells (bottom). **g** Representative multiplex IF images (left) showing the localization of CAR T cells and their spatial relationships

to FAP⁺ stromal cells and EpCAM⁺ tumor cells in syngenetic transplanted 4622 tumors. Quantification is depicted in the adjacent bar graphs (right).
**h** Representative multiplex IF images (left) showing the localization of CAR T cells and their spatial relationships to FAP⁺ stromal cells and PanCK⁺ tumor cells in spontaneous pancreatic tumors in KPC mice. Quantification is depicted in the adjacent bar graphs (right). Data points are mean±SD (*n* = 5 per group) and groups were compared using one-way ANOVA analysis with Dunnett's multiple comparison tests **(c–e, h)** or two-way ANOVA with Tukey's multiple comparisons tests **(b, f, g)**. *p < 0.05, **p < 0.01, ***p < 0.001, and ****p < 0.0001. **b** top panel Day 1: MigR vs. FAP-CAR, p = 0.040; Meso-CAR vs. FAP-CAR, p = 0.026. bottom panel Day 1: MigR vs. FAP-CAR, p = 0.011; Meso-CAR vs. FAP-CAR, p = 0.012. The p values for remaining comparisons are all <0.001 or <0.0001. Source data are provided as a Source Data file.

robust wound-like response under the latter conditions. Therefore, our modified precision-cut tumor slice assay of tumors treated in vivo prior to harvest and harvested at different time-points post-treatment, better reflected the behavior of CAR T cells in the TME in vivo. Using this modified precision-cut tumor slice protocol, we performed systematic real-time 2-photon imaging of the impact of EGFP⁺ CAR T cells treatment on mCerulean⁺ tumor cells, AF647-CD90.2 labeled stromal cells, and matrix architecture based on label-free second harmonic generation (SHG) for fibrillar collagen, to image cell-cell and cell-matrix interactions in real-time in subregions of the TME including, the stromal-rich region surrounding tumor nests and within tumor nests per se.

Consistent with our flow and multiplexed-IF data, only rare GFP⁺ MigR control and Meso-CAR T cells extravasated into the tumor site, even by day 7 post-administration (Fig. 3b, c). Moreover, the few donor T cells detected in these cohorts were restrained by abundant peritumoral stromal cells and fibrillar collagen network, which restricted them from entering tumor nests (Fig. 3b). Furthermore, they exhibited low levels of mean track speed (mean cell migration speed over 20 min), track straightness (ratio of cell displacement to the total length of the trajectory), track length (total length of the trajectory) (Fig. 3d), and few, if any interactions with stromal cells, matrix or tumor cells (Supplementary Movies 1-2) were observed. In contrast, significant numbers of FAP-CAR T cells infiltrated by day 1 and increased at days 4 and 7 post-treatment. Infiltrating FAP-CAR T cells, although highly motile, initially accumulated within the stromal cell and collagen-rich regions surrounding tumor nests being restricted in their mobility by the stroma and prevented from entering tumor nests. Interestingly however, by day 4, and to much greater extent by day 7, after stromal cells and matrix were depleted, FAP-CAR T cells were found escaping from the confines of the stroma and successfully penetrating beyond the tumor borders into tumor nests (Fig. 3b, c and Supplementary Fig. 7b–d). Specifically, FAP-CAR T cells were often observed directionally migrating along remodeled thinner collagen fibers within the tumor nests (Fig. 3d and Supplementary Movies 3-4). Furthermore, FAP-CAR T cells by day 1-4 appeared activated with blast morphology and prominent uropods and were highly motile. These tumor-infiltrating FAP-CAR T cells exhibited relatively high levels of mean track speed and track length (Fig. 3d, e and Supplementary Movies 5-6). The findings based on two-photon imaging were confirmed by the multiplex IF staining of serial sections, further indicating that FAP-CAR T cells can overcome the stromal barrier to infiltrate tumor nests (Supplementary Fig. 7e).

As indicated by flow cytometry, multiplexed-IF, and real-time 2-photon microscopy, stromal cells were reduced to some degree as early as 1 day post-treatment with FAP-CAR T cells. Therefore, to better capture the earliest cell-cell and cell-matrix interactions of FAP-CAR T cells as they entered tumors, we performed similar analyses at earlier time-points. In these studies, we added AF647-anti-CD31 to the slices to image vasculature in addition to PE-CD90.2 to image stromal cells in slices from mCerulean⁺ 4662 tumor-bearing mice at baseline (prior to

FAP-CAR T cell administration) and 6, 12, and 18 hours post FAP-CAR T cell administration. Interestingly, real-time 2-photon imaging and multiplex IF images revealed extravasating GFP⁺ FAP-CAR T cells surrounding peritumoral blood vessels (CD31⁺ or CD31⁺CD90.2⁺) at 6 hours, which was further increased at 12- and 18 hours post-administration of FAP-CAR T cells (Supplementary Fig. 7f, g). Moreover, these FAP-CAR T cells started to crawl toward stromal cells (CD90.2⁺), followed by their arrest in proximity to stromal cells presumably due to stable interactions formed as the result of engagement of their cognate antigen on FAP⁺ stromal cells (Supplementary Fig. 7h and Movies 7-8). These data provide the direct visualization of the role of FAP⁺ stromal cells in mediating immune exclusion, and also provide the evidence that FAP⁺ stromal cells in addition, play a critical role in restricting T cell extravasation.

## Stromal-targeted FAP-CAR T cells rapidly deplete matrix, enhance T cell infiltration and alter the immune and stromal landscapes

To characterize the longer-term impact of disruption of the fibro-proliferative response and matrix remodeling by FAP-CAR T cells, we performed multiplexed IF to spatially profile the stromal and immune landscape alterations in the TME in response to FAP-CAR compared to MigR control T cell or PBS treatments at 1 and 2 weeks post-treatment. Given the known heterogeneous nature of stromal cell subsets in the TME, we stained for FAP, alpha smooth muscle actin [α-SMA], platelet-derived growth factor receptor alpha [PDGFR-α] and/or podoplanin (PDPN). CAFs staining for all these markers were prevalent in tumors treated with PBS or MigR control T cells at 1 week and 2 weeks post-treatment (Fig. 4a, top row). In contrast, FAP⁺-CAFs were selectively depleted in tumors from FAP-CAR T cells treated mice at 1 week post-treatment. However, by 2 weeks post-treatment with FAP-CAR T cells, other CAF subtypes were also depleted (Fig. 4a, top row). The depletion of stromal cells was unexpectedly found to be associated with a rapid and profound loss of matrix, as evidenced by the marked decrease in remodeled collagen as detected by staining with collagen hybridizing peptide (CHP) that reacts with mechanically unfolded collagen molecules, and fibronectin (FN), at both 1 and 2 weeks post-FAP-CAR T cells treatment (Fig. 4a, middle row). Tumors from PBS and MigR treated mice were largely devoid of CD3⁺T cells at both 1 and 2 weeks post-treatment, while the reduction in stromal cells and matrix observed in the tumors from FAP-CAR T cell-treated mice was associated with significant infiltration of CD3⁺ T cells at 1 week that further increased at 2 weeks post-treatment when Pan-CK+ tumor cells were reduced (Fig. 4a, bottom row).

We next analyzed the overall immune profile of tumors by multiparametric flow cytometry (Supplementary Fig. 8). Consistent with the IF data above, flow cytometric analysis of dissociated tumors performed in parallel demonstrated efficient infiltration of FAP-CAR T cells compared with MigR control T cells (Fig. 4b). Endogenous lymphocyte and myeloid cell content in the tumors 1 week post-administration of FAP-CAR, MigR control T cells or PBS were

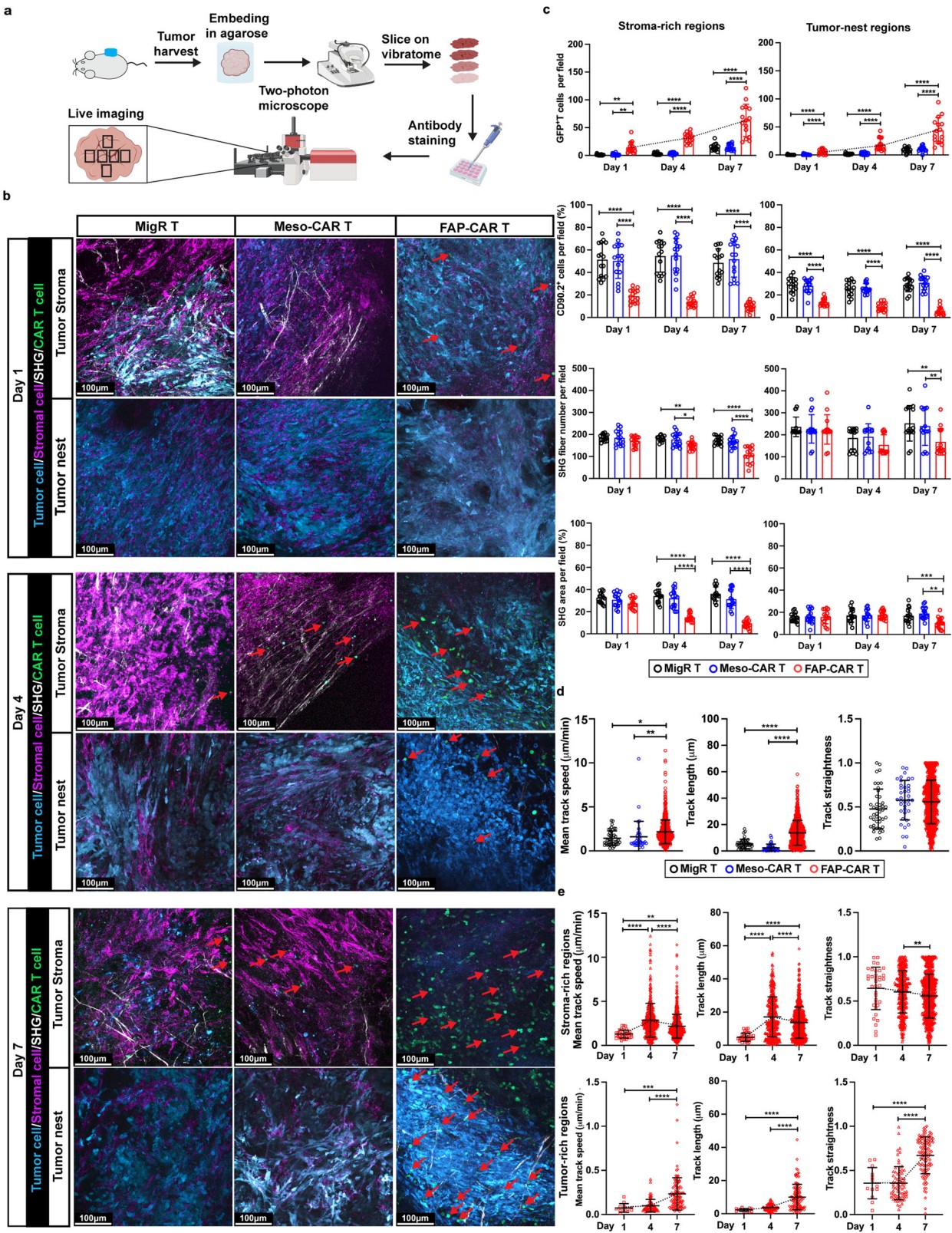

comparable. In contrast, by 2 weeks post-treatment, tumors from FAP-CAR T cells treated mice exhibited greater infiltration/accumulation of endogenous CD3[+] T cells, especially CD8[+] T cells, and NK cells, along with decreased accumulation of granulocyte-like myeloid cells (CD11b[+]Ly6C[lo]Ly6G[+]) (Fig. 4b). Taken together, these data support the idea that FAP-CAR T cells can deplete stromal cells and matrix and substantially modify the immune contexture of the TME by recruiting cytotoxic effector lymphocytes and excluding or killing potentially immunosuppressive myeloid cells.

## Ablation of FAP[+] stromal cells by FAP-CAR T cells enhances the efficacy of immune checkpoint blockade therapy

PD-L1 can be expressed on tumor cells and myeloid cells and, after binding to PD-1 expressed on activated T cells, can induce

**Fig. 3 | FAP-CAR T cells overcome the physical barrier and immunosuppressive TME to infiltrate tumor nests. a** Workflow for two-photon microscopy. **b** Representative two-photon images at subregions of tumors treated with MigR, Meso-CAR, or FAP-CAR T cells at indicated timepoints. GFP⁺ T cells (green, red arrows), stromal cells (magenta), tumor cells (cyan), and SHG (gray). Scale bar, 100 μm. **c** Quantification of GFP⁺ T cells, stromal cells, SHG fiber number, and SHG⁺ area in stroma-rich (left) and tumor nests (right) at indicated timepoints. **d** Quantification of T cell motility (left; mean track speed; middle, track length; right, track straightness) for MigR, Meso-CAR, and FAP-CAR T cells at day 7 post-administration. **e** Quantification of FAP-CAR T cell mobility in stroma-rich (top) and tumor nest (bottom) regions at indicated timepoints. **a** Created with BioRender.com. Data points (15 per condition) in **c** represent measurements in 3 different fields of stroma-rich and 3 different fields of tumor-nest regions from tumor slices from 5 mice per group. Data points in **d, e** represent the quantification of motility of all individual T cells (identified unbiasedly by Imaris) in the 3 most stroma-rich regions and the 3 most tumor cell-rich (nest) regions amongst 6 systematically selected fields for one slice from each of 5 tumors per group; bars, mean ± SD. Groups were compared using two-way ANOVA with Tukey's multiple comparisons tests (**c**) or one-way ANOVA analysis with Dunnett's multiple comparison tests **d, e**. $^*p < 0.05$, $^{**}p < 0.01$, $^{***}p < 0.001$, and $^{****}p < 0.0001$. **c** T cells (left): Day 1 MigR/Meso-CAR vs. FAP-CAR, $p = 0.003$ and 0.004, respectively. Fiber number: left Day 4 MigR/Meso-CAR vs. FAP-CAR, $p = 0.004$ and 0.013, respectively; right Day 7 MigR/Meso-CAR vs. FAP-CAR, $p = 0.001$ and 0.007, respectively. SHG area (right): Day 7 MigR vs. FAP-CAR, $p = 0.004$. **d**. track speed: MigR/Meso-CAR vs. FAP-CAR, $p = 0.002$ and 0.012, respectively. **e** track straightness (left): Day 4 vs. 7, $p = 0.009$. The $p$ values for remaining comparisons are all <0.001 or <0.0001. Source data are provided as a Source Data file.

immunosuppression. Thus, blockade of the PD-L1/PD-1 pathway has been found to enhance anti-tumor immunity and inhibit tumor growth in preclinical studies and in clinical trials for many types of cancers. However, in PDAC, checkpoint blockade has generally been disappointing. This is likely due, at least partially, to the lack of infiltration of effector immune cells in the TME that we validated in the experiments described above. Accordingly, we investigated whether FAP-CAR T cells could enhance the efficacy of treatment with anti-PD-1 in PDAC tumors.

4662 tumor-bearing CD45.2 C57BL/6 mice were treated with either FAP-CAR or MigR control T cells (CD45.1). Four days post-T cell administration, the mice were given isotype control antibody (IgG) or anti-PD-1 antibody twice weekly by intraperitoneal injection. This resulted in 4 experimental groups of mice that received either FAP-CAR T cells and IgG, FAP-CAR T cells and anti-PD-1, MigR control T cells and IgG, or MigR control T cells and anti-PD-1 (Fig. 5a). Consistent with previous reports, anti-PD-1 antibodies had no effect on tumor growth. As above (Fig. 2b), there was significant slowing of the growth of tumors by FAP-CAR T cells. However, the combination of FAP-CAR T cells followed by anti-PD-1 synergistically reduced tumor growth and prolonged survival compared with other treatment combinations (Fig. 5b–e). Flow cytometry and multiplex IF analyses confirmed the depletion of FAP⁺ stromal cells and blockade/neutralization of PD-1 on T cells (Fig. 5f–h). With regard to T cells, we found a significantly greater intra-tumoral accumulation of FAP-CAR T cells when combined with anti-PD-1 (Fig. 5f). Importantly, our data shows that FAP-CAR T cells combined with anti-PD-1 therapy facilitated the recruitment of endogenous CD8⁺ T cells (Fig. 5g–i). Moreover, anti-PD-1 dramatically improved the functionality of cytotoxic CD8⁺ T cells, as evidenced by the down-regulation of co-inhibitory markers such as Tim-3 and the enhancement of TNF-α and IFN-γ expression upon restimulation ex vivo (Fig. 5i). Collectively, these data demonstrate that FAP-CAR T cells enhanced the efficacy of anti-PD-1 therapy in controlling tumor growth.

### FAP-CAR T cells are sufficient to convert a hostile tumor milieu to render PDAC tumors permissive to TAA-CAR T cells and endogenous immune cell infiltration and tumoricidal activity

Although pre-treatment of FAP-CAR T cells followed by anti-PD-1 therapy showed enhanced inhibition of PDAC tumor growth, the overall survival was only modestly improved, thus warranting investigation of other combination therapies. We next investigated whether FAP-CAR T cells can augment anti-tumor immunity of subsequent administration of Meso-CAR T cells. Accordingly, 4662 tumor-bearing syngeneic mice were first treated with MigR, Meso-CAR or FAP-CAR T cells. Two weeks later, mice treated with MigR control T cells received a second dose of MigR control T cells and mice treated with Meso-CAR T cells were given a second treatment with either MigR control or Meso-CAR T cells. In parallel, the mice initially treated with FAP-CAR T cells received a second treatment with either MigR

control, Meso-CAR, or FAP-CAR T cells (Fig. 6a). Dual doses of Meso-CAR T cells had no significant effect on the growth of tumors compared to either dual doses of MigR control T cells or Meso-CAR T cells followed by MigR control T cells. Interestingly, we found that dual doses of FAP-CAR T cells not only slowed growth of the tumors but initially induced a degree of tumor regression compared to tumors in mice that received a first dose of FAP-CAR T cells with a subsequent dose of MigR control T cells. However, most notably, the combination of FAP-CAR and subsequent Meso-CAR T cells showed the strongest inhibitory effect on tumor growth and tumor burden (Fig. 6b, c and Supplementary Fig. 9a–c) and had the greatest effect on survival (Fig. 6d). Importantly, pre-treatment of FAP-CAR T cells followed by Meso-CAR T cells also robustly inhibited and stabilized the tumor growth of autochthonous KPC (Fig. 6e–g) and completely regressed the tumors in AsPC-1 (Fig. 6h–j) and Capan-2 (Supplementary Fig. 9d–f) human PDAC xenograft models, exhibiting high translational potential of our combination strategy in clinic.

To explore mechanistically how FAP-CAR T rendered PDAC tumors sensitive to Meso-CAR T cells, we performed multiplex IF analysis. This analysis demonstrated consistent depletion of CAFs and matrix by FAP-CAR T cells compared to the tumors first treated with MigR control or Meso-CAR T cells (Supplementary Fig. 10a, b). Moreover, a higher number of total T cells accumulated in tumor nests accompanied by significantly decreased expression of PanCK⁺ tumor cells and depletion of PDPN⁺ CAFs in the tumors treated with FAP-CAR T cells followed by FAP-CAR and Meso-CAR T cells than in other combinations (Supplementary Figs. 10a, b). Notably, FAP-CAR T cells combined with subsequent Meso-CAR T cells significantly increased apoptosis and suppressed proliferation of tumor cells, as evidenced by up-regulated cleaved caspase-3 and down-regulated Ki-67 on tumor cells (Fig. 6k). Furthermore, we found that pre-treatment with FAP-CAR T cells reduced the proportion of TGF-β producing cells and intra-tumoral angiogenesis, especially when combined with subsequent administration of a second dose of FAP-CAR T cells or Meso-CAR T cells (Fig. 6k). Flow cytometry analyses of tumors at the endpoint further confirmed the reduction in TGF-β⁺ cells, which reflected the decrease in the number of α-SMA⁺ CAFs and myeloid cells, the main sources of TGF-β secretion (Supplementary Fig. 10c).

To test our hypothesis that FAP-CAR T cells can enhance the trafficking, infiltration, and functionality of subsequently administered Meso-CAR T cells, CD45.2 C57BL/6 mice were utilized as donors to produce EGFP-tagged MigR control, Meso-CAR, and FAP-CAR T cells, and these T cells were adoptively transferred via intravenous injection into CD45.1 C57BL/6 mCerulean labelled PDAC tumor-bearing hosts. 4 days post-administration of the first treatment, these mice were treated with tdTomato-tagged Meso-CAR T cells which were derived from tdTomato mice and transduced to express mouse Meso-CAR without the GFP fluorescence tag as the second treatment. 2-photon microscopy, multiparametric flow cytometry, and multiplexed IF were performed at day 1, 4 and 7 post-second treatment (Fig. 7a). Under

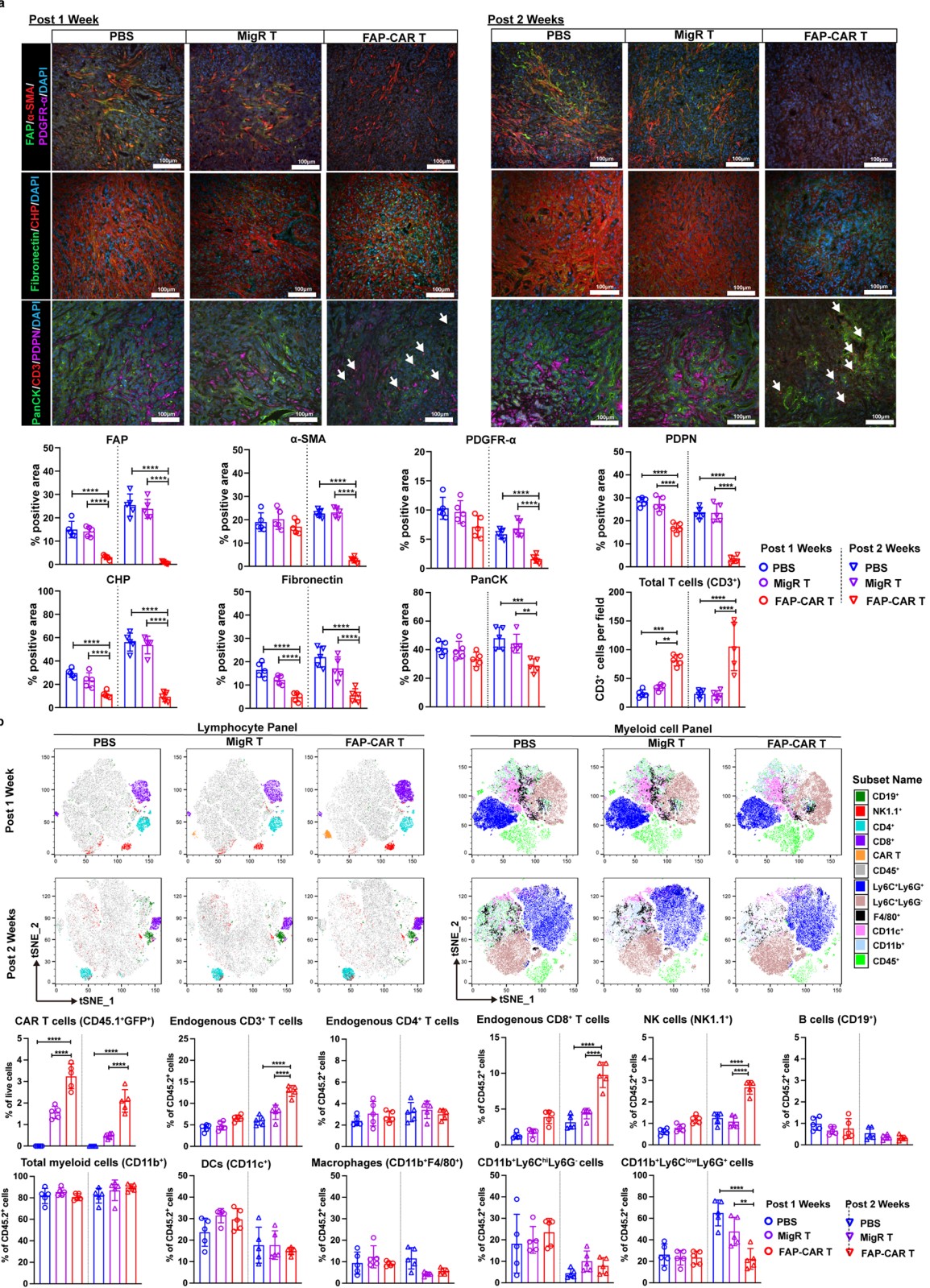

these conditions, exogenous (CD45.2) and endogenous (CD45.1) T cells, as well as the first (EGFP) and second (tdTomato) dose of CAR T cells could be distinguished and characterized. Consistent with our previous results described above, we found significantly more GFP+ FAP-CAR T cells accumulated in both stromal cell-rich and tumor nest regions at day 1, 4 and 7 post-second treatment compared with GFP+ MigR control and Meso-CAR T cells, resulting in the rapid ablation of

FAP+ stromal cells and mesothelin+ tumors cells (Fig. 7b, c and Supplementary Fig. 11a, b). Dramatically increased numbers of tdTomato+ Meso-CAR T cells started to infiltrate into both tumor stroma and tumor nest regions, even at day 1 post-second treatment in tumors first treated with FAP-CAR T cells compared with the tumors first treated with MigR control or Meso-CAR T cells (Fig. 7b, c). Moreover, these tdTomato+ Meso-CAR T cells infiltrated and accumulated in tumor

**Fig. 4 | Ablation of FAP⁺ cells alters the stromal landscape, rapidly depletes matrix and enhances T cell infiltration. a** Representative (top) and quantification of (bottom) multiplex immunofluorescence images in tumors 1 week (left) and 2 weeks (right) post-treatment with PBS, MigR control or FAP-CAR T cells. First row depicts staining for stromal cell profiles, including FAP (green), α-SMA (red) and PDGFR-α (magenta). Second row depicts staining for ECM components including collagen hybridizing peptide (CHP) that detects remodeled collagen (red) and FN (green). Third row depicts distribution of T cells (CD3, red) and their spatial relationships to tumor cells (PanCK, green) and stromal cells (PDPN, magenta). Nuclei were stained with DAPI (blue). Scale bar: 100 μm. **b** t-SNE of multi-parametric flow cytometry demonstrating the immune cell profiles based on the gating strategy as depicted in Supplementary Fig. 8, including lymphocyte (left) and myeloid cells (right) in tumors 1 week (Top row) and 2 weeks (bottom row) post-treatment with PBS, MigR control or FAP-CAR T cells. The percentages of indicated immune cell subpopulations in total live cells or CD45.2⁺ cells from tumors for each group of mice quantified based on multi-parametric flow cytometry analysis. Data points are mean ± SD ($n = 5$ per group) and groups were compared using two-way ANOVA with Tukey's multiple comparisons tests. $^*p < 0.05$, $^{**}p < 0.01$, $^{***}p < 0.001$, and $^{****}p < 0.0001$. **a.** PDGFRα (post 1 week): PBS vs. FAP-CAR, $p = 0.032$. PDGFRα (post 2 weeks): PBS vs. FAP-CAR, $p = 0.002$. Fibronectin (post 1 week): MigR vs. FAP-CAR, $p = 0.022$. CD3 (post 1 week): MigR vs. FAP-CAR, $p = 0.005$. PanCK (post 2 weeks): MigR vs. FAP-CAR, $p = 0.004$. **b** post 1 week: CD3 PBS vs. FAP-CAR, $p = 0.047$. CD8 PBS vs. FAP-CAR, $p = 0.004$, MigR vs. FAP-CAR, $p = 0.020$. post 2 weeks: Ly6C^low Ly6G⁺ MigR vs. FAP-CAR, $p = 0.006$. The p values for remaining comparisons are all <0.001 or <0.0001. Source data are provided as a Source Data file.

nests in a time-dependent manner with profound depletion of mCerulean⁺ tumor cells over time (Fig. 7b, c and Supplementary Movies 9-10). Real-time 2-photon microscopy confirmed that the first treatment of FAP-CAR T cells rendered tumors permissive to Meso-CAR T cells allowing their extravasation, infiltration of and function within tumor nests. Flow cytometry and multiplex IF analyses further confirmed that the number of CD45.2⁺tdTomato⁺ Meso-CAR T cells in tumors first treated with FAP-CAR T cells were significantly higher than those treated with MigR control and Meso-CAR T cells (Fig. 7d and Supplementary Fig. 11c, d). Also important was the observation that a first dose of FAP-CAR T cells enhanced the activation of CD45.2⁺tdTomato⁺ Meso-CAR T cells in tumor sites, as demonstrated by increased expression of GzmB, TNF-α, and IFN-γ after stimulation (Fig. 7e). Collectively, these data suggest that ablation of FAP⁺-CAFs by FAP-CAR T cells have translational potential for enhancing the efficacy of Meso-CAR T cell therapy, and likely other TAA targeted therapies, in solid tumors.

### Sequential administration of FAP-CAR T cells with Meso-CAR T cells enhances systemic endogenous adaptive anti-tumor immunity in PDAC models

The data above indicates that the combination of FAP- and Meso-CAR T cells enhanced the anti-tumor activity of Meso-CAR T cells. Moreover, we found FAP-CAR T cells enhanced the anti-tumor activity of anti-PD-1 suggesting that FAP-CAR T cells alone may have an inherent capacity to modulate the endogenous immune landscape. Therefore, to address if the changes in the TME induced by the combination of FAP- and Meso-CAR T might have the potential to engage endogenous anti-tumor immunity, 4662 tumor-bearing C57BL/6 were pre-treated with MigR control or FAP-CAR T cells. 2 weeks later, the mice receiving MigR control T cells were treated with a second dose of MigR control T cells, whereas the mice given first dose of FAP-CAR T cells received a second treatment of MigR control, FAP-CAR, or Meso-CAR T cells (Fig. 8a). Consistent with our data above (Fig. 6b), we found pre-treatment of FAP-CAR T cells significantly enhanced the efficacy of the second treatment with Meso-CAR T cells in inhibiting tumor growth (Supplementary Fig. 12a). Based on flow cytometry, we found that FAP-CAR T cells followed by subsequent Meso-CAR T cells treatment increased the infiltration of endogenous CD8⁺ T cells in the tumor sites and a concomitant decrease in CD4⁺FoxP3⁺ Treg cells (Fig. 8b and Supplementary Fig. 12b, c) and increase in IFN-γ expression in endogenous CD8⁺ T cells after restimulation compared to other combinations (Fig. 8c). The combination of FAP-CAR and Meso-CAR T cells also led to the strongest activation of endogenous splenic CD8⁺ T cells, as demonstrated by the high levels of expression of IFN-γ after restimulation (Fig. 8c). Moreover, in addition to the decreased levels of Ly-6C⁺Ly-6G⁺ myeloid cells and F4/80⁺CD206⁺ tumor-associated macrophages (TAMs) (Fig. 8d and Supplementary Fig. 12d–f), we found that the rare conventional dendritic type 1 cells (cDC1, CD45⁺CD103⁺CD11c⁺) subpopulation, which plays a central role in the adaptive immune response by cross-presenting antigen to cytotoxic CD8⁺ T cells, was significantly increased in the tumors treated with FAP-CAR T cells and Meso-CAR T cells (Fig. 8e), indicating that the mobilization of endogenous CD8⁺ T cells might be facilitated by the presence of cross-presenting cDC1s. Multiplexed IF analyses also validated the increase in endogenous CD8⁺ T cells (especially IFN-γ⁺CD8⁺ T cells) and cDC1, and indicated that potentially immune suppressive cells, such as TAMs (especially CD206⁺ M2 macrophages) and Treg cells were excluded (Fig. 8f). Collectively, these data indicate that the immune-desert/excluded TME of PDAC was reprogrammed by the combination of FAP-CAR and Meso-CAR T cells. Together, these results suggest that the combination of FAP-CAR and Meso-CAR T cells treatment can also promote endogenous adaptive anti-tumor immunity.

## Discussion

Whereas tumor antigen-targeted (TAA) CAR T cells have shown remarkable success in treating some hematopoietic cancers, extending this innovative immunotherapeutic approach to the treatment of solid tumors has proven challenging, at least partially due to limited tumor infiltration, penetration, and loss of functionality within the solid tumor microenvironment. However, in-roads have been made in pre-clinical models of solid tumors using stromal cell-targeted FAP-CAR T cells in multiple tumor types[9,11]. Notably, FAP-CAR T cells have shown significant efficacy even in the highly desmoplastic KPC syngeneic transplant and autochthonous GEMM models of PDAC[9,11] in which the tumor antigen mesothelin-targeted CAR T cells have had very modest, if any effect[24,26,33]. Herein, we demonstrate that FAP⁺ stromal cells and associated matrix minimize T cell recruitment and mediate immune exclusion and immunosuppression. By ablating FAP⁺ stromal cells and matrix, FAP-CAR T cells overcome these barriers and thereby successfully extravasate into tumors, eventually infiltrate into tumor nests, and retain their functionality. Moreover, we demonstrate that these effects of FAP-CAR T cells are sufficient to render tumors permissive to TAA-CAR T cells and endogenous immune cell infiltration and tumoricidal activity, resulting in enhanced anti-tumor activity of Meso-CAR T cells and anti-PD-1 when given in combination with FAP-CAR T cells in our pre-clinical models of PDAC tumors, and as previously reported, a tumor vaccine in combination with FAP-CAR T cells in a more immunogenic mouse model of lung cancer[11]. These results provide a strong rationale for translating combinations of stromal cell-targeted and TAA-targeted therapies or ICT to patients.

This study provides mechanistic insights into at least three of the roles of stromal cells and matrix in severely limiting the impact of various tumor cell-directed immunotherapies in the context of solid tumors. First, FAP-CAR T cells overcame restrictions to T cell extravasation and perivascular invasion. This is consistent with the presence of FAP⁺ perivascular cells that are eliminated by FAP-CAR T cells, thereby reversing the impediment to FAP-CAR T cell infiltration and to Meso-CAR T cells when given in combination[34].

Second, we found that FAP-CAR T cells disrupted immune-exclusion by depleting the stromal cells and matrix-dense network

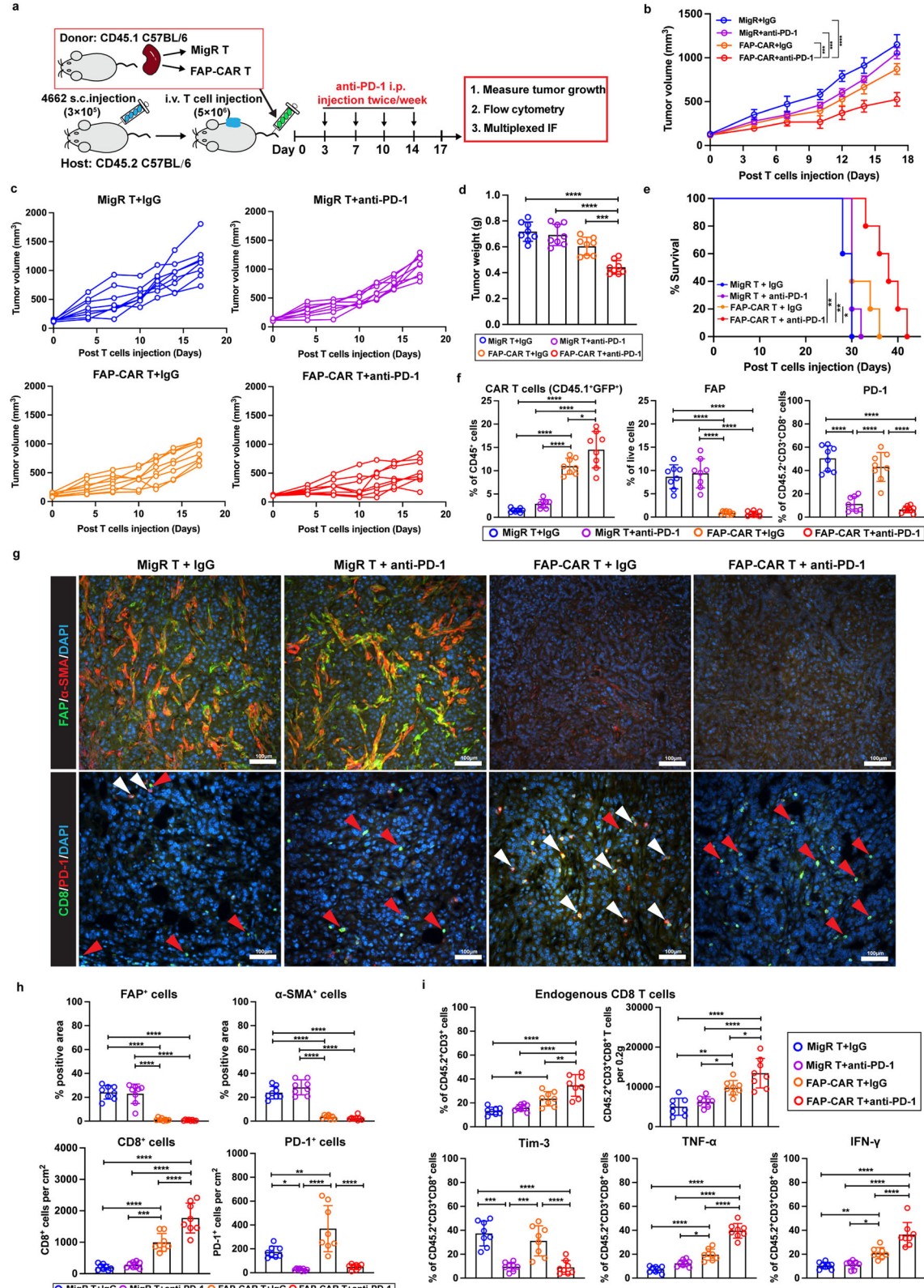

surrounding tumor nests that deter T cell penetration of the stromal border and prevent their direct contact with tumor cells. As the stromal border was found to be rich in FAP⁺ cells[17], this is likely due to the ablation of these FAP⁺-CAFs that elaborate and align fibrillar matrix. Loss of targeted stromal cells at the tumor border was notably associated with nearly complete clearance of fibrillar collagen. This is certainly due in part to depletion of stromal cells responsible for

depositing matrix and possibly, also due to increases in the activity of remodeling enzymes. However, the dramatic and rapid kinetics of the loss of matrix leads us to speculate that stromal cells, through their impact on matrix architecture and the physical tension they impose on matrix fibrils, may also protect against proteolytic degradation and turnover of matrix. Future studies will address this hypothesis. In any case, disruption of the stromal network was associated with

**Fig. 5 | Ablation of FAP+ cells by FAP-CAR T cells enhances the efficacy of anti-PD-1 antibody immune checkpoint therapy. a** Treatment protocol: 4662 tumor-bearing CD45.2 C57BL/6 mice were treated with MigR control or FAP-CAR T cells followed by isotype or anti-PD-1 antibodies. **b** Average tumor growth curves, **c** individual tumor growth curves, **d** end-point tumor weights, and **e** modified Kaplan-Meier curve for mice in each of the indicated treatment groups. **f** Quantification of tumor-infiltrating MigR control or FAP-CAR T cells expressing both CD45.1 and EGFP, FAP+ cells and PD-1 expression in endogenous CD8+ T cells in tumors at end-point. **g** Representative multiplex IF images and **h** quantification of FAP (green) and α-SMA (red) (top row), as well as CD8 (green) and PD-1 (red) (bottom row) in end-point tumors for indicated treatment groups. Nuclei were stained with DAPI (blue). Scale bar, 100 μm. **i** Quantification of tumor infiltrating endogenous CD8+ T cells and total Tim-3+, TNF-α+, and IFN-γ+ T cells upon re-stimulation ex vivo based on flow cytometry analysis. Data points are mean ± SD

($n$ = 8 per group) and groups were compared using one-way ANOVA analysis with Dunnett's multiple comparison tests (**b, d, f, h**, and **i**) or log-rank test (**e**). *$p < 0.05$, **$p < 0.01$, ***$p < 0.001$, and ****$p < 0.0001$. **e** MigR+IgG/MigR+anti-PD-1/ FAP-CAR+IgG vs. FAP-CAR+anti-PD-1, $p = 0.004$, 0.003 and 0.028, respectively. **f** CAR T cells: FAP-CAR+IgG vs. FAP-CAR+anti-PD-1, $p = 0.015$. **g** PD-1+ cells: MigR+IgG vs. MigR+Anti-PD-1, $p = 0.030$; MigR+IgG vs. FAP-CAR+IgG, $p = 0.003$. **i** upper left panel: MigR+IgG vs. FAP-CAR+IgG, $p = 0.008$; FAP+IgG vs. FAP-CAR+anti-PD-1, $p = 0.004$. upper right panel: MigR+IgG vs. FAP-CAR+IgG, $p = 0.003$; MigR+anti-PD-1 vs. FAP-CAR+IgG, $p = 0.032$; FAP-CAR+IgG vs. FAP-CAR+anti-PD-1, $p = 0.025$. Bottom panel: TNF-α MigR+anti-PD-1 vs. FAP-CAR+IgG, $p = 0.012$; IFN-γ MigR+IgG vs. FAP-CAR+IgG, $p = 0.009$; MigR+anti-PD-1 vs. FAP-CAR+IgG, $p = 0.015$. The $p$ values for remaining comparisons are all <0.001 or <0.0001. Source data are provided as a Source Data file.

resumption of FAP-CAR T cell motility and a significant number of FAP-CAR T cells successfully traversing the tumor border and entering into the tumor nests where they were often observed directionally migrating along thinner elongated collagen fibers within the tumor nests. In a similar manner, FAP-CAR T cells enhanced access of Meso-CAR T cells to tumor nests when given in combination.

Although beyond the scope of this study, it will be of interest going forward to investigate additional stromal-dependent, immune-independent, pro-tumorigenic axes that may also be disrupted by FAP-CAR T cells. For example, CAFs can produce hepatocyte growth factor (HGF), which further activates its cognate receptor, c-Met, on tumor cells resulting in a pro-tumorigenic environment by triggering invasive and metastatic behavior of tumor cells[35,36]. As such, the broad depletion of stromal cells by FAP-CAR T cells may induce the reduction of tumor-enhancing growth factors including HGF and c-Met, thus limiting the proliferation and invasiveness of cancer cells.

Third, FAP-CAR T cells reprogrammed the immunosuppressive milieu such that they retained their functionality and also prevented loss of function of subsequently administered Meso-CAR T cells and tumor-infiltrating endogenous T cells when given in combination with anti-PD1. Combinatorial therapies with TAA-CAR T cell therapies and ICT are being pursued in the clinic. However, such combinations are predicted only to be successful if sufficient numbers of TAA-CAR T cells penetrate surrounding stroma to gain access to their target tumor cells; to date, ICT has proven to have little impact on CAR T cell-mediated tumor growth control[37]. In contrast, we found that pre-treatment with stromal cell-targeted FAP-CAR T cells had additive to synergistic effects when followed by PD-1 blockade. Although our results, together with previous studies that showed depletion of FAP+-CAFs enhanced the efficacy of anti-cytotoxic T lymphocyte-associated protein 4 (CTLA-4) and programmed cell death ligand 1 (PD-L1) blockade[8] and prolonged survival, the response was transient with resumption of tumor growth and demise of all animals in time. Promisingly, we found a combination of FAP-CAR T cells followed by Meso-CAR T cells had a more robust effect with tumor stabilization and even regression observed in some PDAC-bearing mice that translated into greater prolongation of survival. In a single other study of a nascent lung metastasis model, consistent with the studies herein, Kakarla et al.[16] found that combining FAP-specific T cells with T cells that targeted the EphA2 antigen on A549 cancer cells, enhanced overall anti-tumor activity and conferred a survival advantage compared to either alone. However, whereas we treated mice with established tumors, in their study A549 tumor cells were intravenously injected and then received treatment just 4 days post-tumor cell inoculation, prior to generation of established tumors.

The synergy we observed between FAP-CAR T cells given in combination with Meso-CAR T cells or anti-PD-1 can, at least in part, be explained by changes in the immune landscape of the tumor microenvironment, as well as their impact on systemic immunity. Specifically, we demonstrate that treatment with FAP-CAR T cells resulted in increased recruitment of cytotoxic effectors (CD8+ T and NK cells) and decreased the number of immunosuppressive myeloid cells, which was associated with down-regulated TGF-β signaling. FAP+-CAFs also secrete the CXC-chemokine ligand 12 (CXCL12) that hinders T cell recruitment and secrete the chemokine ligands CCL2, CCL3, CCL4, and CCL5[8,38] that recruit myeloid cells. Thus, depletion of FAP+-CAFs by FAP-CAR T cells can also promote intra-tumoral infiltration of T cells by impairing the CXCL12-CXCR4 axis and down-regulating chemokines that influence myeloid cells recruitment into the TME[38]. Moreover, as CAFs reportedly can compete metabolically with T cells, FAP-CAR T cells also likely alter the metabolic status of the TME in favor of endogenous anti-tumor immunity and functionality of adoptively transferred CAR T cells[39]. Of interest, we also observed systemic changes in immune response in CD8 and type I DCs, which might result either from reduced tumor burden and or from the induced release of cytokines or other factors by the activated CAR T cells. Future studies will determine the contributions of these potential mechanisms.

It is important to note that CAFs are heterogeneous, with FAP expressed on a prominent subset, but not all CAFs. Another well-characterized subset of CAFs, referred to as myCAFs, are defined by their expression of αSMA+. These FAP+ and αSMA+ subsets of CAFs are phenotypically and functionally distinct but overlapping, with PDAC tumors also containing a population of double positive FAP+αSMA+ CAFs[9,40–44]. Expectedly, and consistent with prior reports, we found that FAP+ cells were initially selectively depleted up to one-week post-administration of FAP-CAR T cells with little impact on prevalence of αSMA+ cells. However, a significant depletion of αSMA+ CAFs became evident by two weeks post-treatment with FAP-CAR T cells. There are several, not mutually exclusive, explanations for this observation, including, that αSMA+ cells may derive from FAP+ progenitors that once depleted are no longer available as a reservoir to maintain the αSMA+ population. This is, however, unlikely the sole explanation, since we noted that FAP+ cells can be replenished over time[9]. Another possibility is that the majority of αSMA+ cells that are eventually depleted also express lower levels of FAP and that loss of FAPhiαSMAlo cells are depleted more readily and earlier, followed over time by depletion of FAPloαSMAhi CAFs. In addition, as previously posited[45,46], FAP+ cells may mediate early matrix remodeling leading to the generation of a collagen-rich and increasingly stiff matrix that is required for the generation of αSMA+ CAFs, such that the loss of FAP+ cells leads to modifications of matrix that can no longer sustain αSMA+ CAFs. We further demonstrated that the level of TGF-β, which is critical in translating signals from ECM and/or stiffness to initiate and facilitate the fibrotic process and immunosuppression[47,48], was down-regulated by FAP-CAR T cells administration, partially supporting the broad remodeling of fibrotic TME, including stromal cells, ECM and even stiffness.

An important translational issue to consider for FAP-CAR T therapy is that murine data showed that FAP was expressed at low levels in some healthy tissues including bone marrow mesenchymal stem cells,

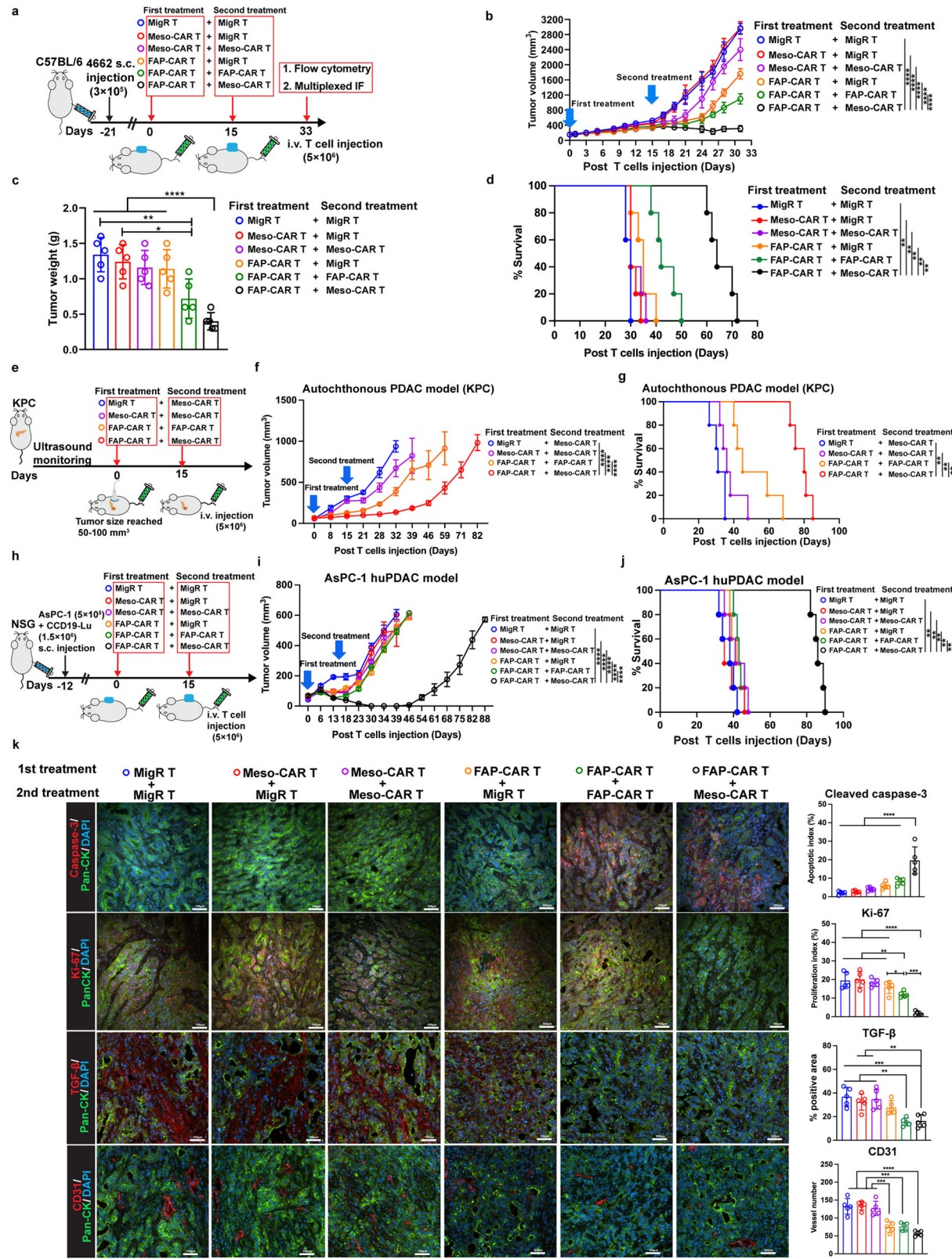

muscle, and the pancreas[49,50]. Complete genetic ablation of FAP-positive cells caused some toxicity (primarily in the bone marrow) that was also seen using one specific FAP-CAR construct[51]. However, our data suggest that a therapeutic window will be present. In our hands, using a different murine-specific FAP-CAR T, and in other studies[16,52,53], no serious adverse effects were observed. A human biodistribution study with [18F]AlF-FAPI-74 demonstrated low uptake in normal organs[54], providing further support that FAP expression is limited in healthy adult tissues. In our recently published study using a single dose of the 4G5 FAP-CARTs, we saw no obvious toxicity[27]. As described above (Supplementary Fig. 3), when we conducted formal autopsy studies using two doses of the 4G5 FAP-CAR T cells, we saw reversible mononuclear infiltration of muscle, pancreas, and lung, but with no changes in bone marrow, nor in anemia markers in blood or animal

**Fig. 6 | Ablation of FAP⁺-CAFs by FAP-CAR T cells enhances the efficacy of subsequent treatment with TAA Meso-CAR T cells in PDAC tumors. a** Treatment protocol: C57BL/6 mice bearing established 4662 tumors were treated with the indicated first dose of T cells and 15 days later treated with the second dose of indicated T cells. **b** Average growth curves, **c** tumor weights at end-point and **d** modified Kaplan–Meier curves of mice from each of the indicated treatment groups. **e** Treatment protocol in established KPC mice. **f** Average growth curves and **g** modified Kaplan–Meier curves of KPC mice from each of the indicated treatment groups. **h** Treatment protocol in NSG mice bearing AsPC-1 human PDAC tumors. **i** Average growth curves and **j** modified Kaplan–Meier curves of AsPC-1 human PDAC bearing NSG mice from each of the indicated treatment groups. **k** Representative multiplexed IF images (top panels) and quantification (bottom panels) of sections of end-point tumors from each of the indicated treatment groups stained for cleaved caspase-3, Ki-67, TGF-β or CD31 (red) as indicated and in each case co-stained with Pan-CK (green). Nuclei were stained with DAPI (blue).

Scale bar, 100 µm. Data points are mean±SD (n = 5 per group) and groups were compared using one-way ANOVA analysis with Dunnett's multiple comparison tests (**b, c, f, i, k**) or log-rank test (**d, g, j**). **e** The pancreas and ultrasound are created with BioRender.com. $^*p < 0.05$, $^{**}p < 0.01$, $^{***}p < 0.001$, and $^{****}p < 0.0001$. **c** MigR+MigR/Meso-CAR+MigR vs. FAP-CAR + FAP-CAR, p = 0.005 and 0.022, respectively. **d** MigR+MigR/Meso-CAR+MigR/Meso-CAR+Meso-CAR/FAP-CAR+MigR/FAP-CAR + FAP-CAR vs. FAP-CAR+Meso-CAR, $p = 0.003, 0.002, 0.002, 0.002$ and $0.002$, respectively. **g** MigR+Meso-CAR/Meso-CAR+Meso-CAR/FAP-CAR + FAP-CAR vs. FAP-CAR+Meso-CAR, p = 0.003, 0.002 and 0.002, respectively. **j** $p = 0.002$ for all comparisons. **k** Ki-67: MigR+MigR/Meso-CAR+MigR/Meso-CAR+Meso-CAR vs. FAP-CAR + FAP-CAR, $p = 0.008, 0.004$ and $0.032$, respectively. TGF-β: Meso-CAR+MigR/Meso-CAR+Meso-CAR vs. FAP-CAR+Meso-CAR, $p = 0.008$ and $0.002$, respectively. Meso-CAR+MigR/Meso-CAR+Meso-CAR vs. FAP-CAR + FAP-CAR, $p = 0.005$ and $0.001$, respectively. The $p$ values for remaining comparisons are all <0.001 or <0.0001. Source data are provided as a Source Data file.

weights compared to the empty vector control. Although a cautious dose escalation strategy will need to be taken, as there may be some patients who may be more likely to experience on-target, off-tumor toxicity (for example in the case of an unknown fibrotic process), our data to date suggest that FAP-CAR T cell therapy will be safe.

In conclusion, our results provide promising data to support the development of combinatorial approaches including stromal cell and tumor antigen-targeted immunotherapies for translation to the clinic in the setting of highly desmoplastic cancers (Supplementary Fig. 13).

## Methods
### Animals
All animal experiments were approved by the Institutional Animal Care and Use Committee (IACUC) of the University of Pennsylvania (Animal Welfare Assurance no.: D16-00045 [A3079-07], Protocol no.: 805003 and 805004) and were carried out in accordance with the guidelines. C57BL/6 J (CD45.1 donor and CD45.2 recipient), NOD/SCID/IL2-receptor γ chain knockout (NSG) and B6.129(Cg)-*Gt(ROSA) 26Sor* ^tm4(ACTB-tdTomato,-EGFP)Luo^/J (mT/mG) mice were purchased from Jackson Laboratory. Fully backcrossed *Kras^G12D^;Trp53^R172H^;Pdx-1-Cre* (KPC) C57BL/6 mice were bred and maintained in our institution. Mice were housed in a specific pathogen-free (SPF) condition at ambient temperature (22 ± 2 °C), air humidity 40–70% and 12 h dark/12 h light cycle and had free access to water and chow. Animal health status was routinely checked by qualified veterinarians.

For the transplant models, tumor cells were inoculated into female mice aged 8-10 weeks of similar weight, randomized before tumor inoculation. KPC mice were enrolled based on the size of spontaneously arising tumors rather than age or weight. All animal experiments were performed in the same well-controlled pathogen-free facility and fed ad libitum. Since the cell line (4662) for the syngeneic transplant model was derived from a tumor from a female KPC mouse, only female mice hosts were utilized. To be consistent, only female mice were used for the xenograft models. However, male and female mice were included in the studies of the genetically engineered autochthonous KPC model. As based on two-side Student's *t* tests, tumor growth was comparable in male and female KPC mice, the data from both sexes were analyzed together. No tumors exceeded the maximum diameter of 2 cm approved by IACUC at the time the mice were sacrificed for tumor analysis or in the modified survival studies.

### Cell lines
Human CCD-19 lung fibroblast (CCD19-Lu), human AsPC-1 and Capan-2 PDAC tumor cells, human 293T cells and Phoenix-ECO packaging cells were purchased from ATCC. Mesothelin-positive 4662 cells were established from the fully backcrossed C57BL/6J KPC PDAC mouse model and kindly provided by Dr. Robert H. Vonderheide (University of Pennsylvania)[9]. 3T3.mFAP.GFP/Luc expressing high level of FAP and 3T3.GFP/Luc without FAP expression cells were generated by

transduction with the lentivirus of firefly luciferase as previously described[11]. All cell lines were tested for the presence of mycoplasma contamination (MycoAlert Mycoplasma Detection Kit, Lonza). For Cerulean labeling, 4662 PDAC cells were transduced with mCerulean retrovirus (Addgene# 96936)[29] to generate mCerulean⁺ 4662 PDAC cells and mCerulean^hi cells were sorted. Mesothelin knock-out 4662 cell line was generated by using *CRISPR-Cas9* system. Briefly, the gRNA oligonucleotides against murine mesothelin (5′- ATGTGGATG-TACTCCCACGG-3′) (synthesized by Dr. Genewiz, MA, USA) were cloned into lentiCRISPRv2 hygro vector (Addgene# 98291) as previously reported[55]. The plasmids were then packaged into lentiviral particles using 293 T cells. 4662 cells were infected with lentivirus and selected by 250 µg/ml Hygromycin B for a week. Single cell clones were isolated using limited dilution and finally identified by western blot and flow cytometry. All cancer cells were cultured at 37 °C with 5% $CO_2$ in DMEM including 10% heat-inactivated Fetal Bovine Serum (FBS), 100 U/ml penicillin-streptomycin and L-glutamine. Phoenix packaging cells (ATCC) were maintained at 37 °C with 5% $CO_2$ in RPMI-1640 including 10% heat-inactivated FBS, 50 U/ml penicillin-streptomycin and L-glutamine. CCD-19 lung fibroblasts were cultured at 37 °C with 5% CO2 in MEM including 10% heat-inactivated FBS, sodium pyruvate, non-essential amino acids, 50 U/ml penicillin-streptomycin and L-glutamine.

### Method Details
**Vector generation, T cell isolation and expansion, and generation of CAR T cells.** The second-generation anti-mouse mesothelin A03 CAR retroviral construct, which included anti-mouse mesothelin scFv, mouse CD8α hinge, CD8α transmembrane domain, mouse 4-1BB costimulatory domain, mouse CD3ζ signaling domain and EGFP reporter, was generated as previously reported[24]. The anti-human mesothelin M11 scFv was generated from a human phage display library and selected for its ability to bind to purified human mesothelin. The VH and VL variable domains of the M11 scFv were fused with a human CD8α hinge, CD8α transmembrane domain, and two cytoplasmic domains derived from 4-1BB and CD3ζ. This anti-human Meso-CAR M11 was subcloned into the MigR1 retroviral vector, which also expresses EGFP, using an internal ribosomal entry site (IRES) as we have previously describe[56]. The second-generation anti-FAP CAR construct was generated by cloning the 4G5 scFv followed by human CD8α hinge, CD8α transmembrane domain, 4-1BB costimulatory domain and CD3ζ signaling domain into the MigR1 retroviral backbone, upstream of IRES and EGFP[27]. This CAR targets both human and murine FAP.

Mouse T cells were isolated from the spleen with T cell isolation kit (STEMCELL) according to the manufacturer's instructions. T cells were stimulated with Dynabeads Mouse T-Activator CD3/CD28 beads (Gibco) for 48 hours and were separated from beads prior to transduction. Infective particles were generated from the supernatants of

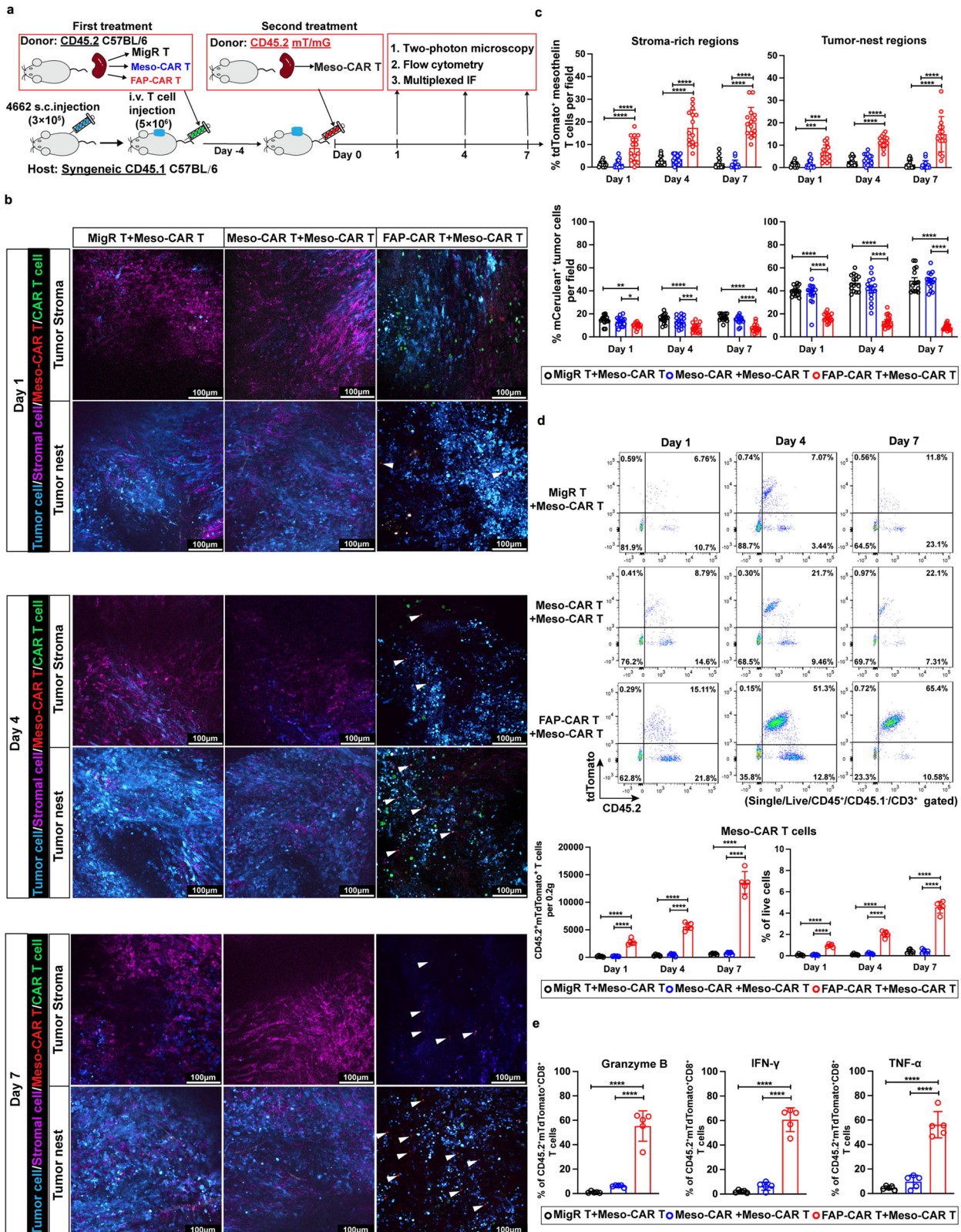

Phoenix-Eco cells transfected with the retroviral vector plasmid and helper plasmids using Lipofectamine 2000 (Invitrogen) as we have previously described[9,11,57]. Indicated retroviruses were added into RetroNectin (20 mg/mL, Takara) pre-coated untreated 24-well plate and spun down at 1000 g for 1 hour. Activated T cells were transduced with retroviruses at MOI of 3000 for 48 hours. Transduction efficiency was calculated based on expression of anti-mesothelin or anti-FAP CAR,

assessed by staining with APC F(ab'$_2$) fragment goat anti-human (Meso-CAR) or mouse (FAP-CAR) IgG (Jackson ImmunoResearch) respectively, and EGFP expression using flow cytometry.

**Cytotoxicity and cytokine release assay.** For checking the cytotoxicity of FAP-CAR T cells in vitro, 3T3.GFP/Luc and 3T3.mFAP.GFP/Luc cells were co-cultured with untransduced (UTD), MigR or FAP-CAR

**Fig. 7 | FAP-CAR T cells rendered tumors permissive to Meso-CAR T cells allowing their infiltration and function within tumor. a** Treatment protocol: 4662 tumor-bearing CD45.1 C57BL/6 mice (hosts) were treated with donor MigR control, Meso-CAR or FAP-CAR T cells (CD45.2 GFP⁺). 4 days later, all mice received a second treatment of CD45.2 tdTomato⁺ Meso-CAR T cells. **b** Representative static images from two-photon microscopy at different subregions in the tumors treated with MigR control, Meso-CAR, or FAP-CAR T cells (GFP⁺) with subsequent Meso-CAR T cells (tdTomato⁺) at indicated time points. The GFP⁺ T cells from first treatment appear green and tdTomato⁺ Meso-CAR T cells from second treatment appear red and examples of latter are highlighted by white arrows. CD90.2⁺ stromal cells appear magenta and mCerulean⁺ tumor cells cyan. Scale bar, 100 μm. **c** Quantification of tdTomato⁺ Meso-CAR T cells (Top panels) and mCerulean⁺ tumor cells (bottom panels) in stroma-rich (left) and tumor nest (right) regions at day 1, 4 and 7 post administration. Data points (15 per condition) represent

measurements in 3 different fields of stroma-rich and 3 different fields of tumor-nest regions from tumor slices from 5 mice per group. **d** Representative flow plots and quantification of flow cytometry analysis of tdTomato⁺ Meso-CAR T cells in tumors from the indicated treatment groups at the indicated times post-second treatment. **e** Quantification of the expression of GzmB, IFN-γ and TNF-α by examined for the infiltrating tdTomato⁺ Meso-CAR T cells in tumors from each of the indicated treatment groups at the indicated times post-second treatment. Data points are mean±SD ($n = 5$ per group) and groups were compared using two-way ANOVA with Tukey's multiple comparisons tests **(c, d)** or one-way ANOVA analysis with Dunnett's multiple comparison tests **(e)**. $^*p < 0.05$, $^{**}p < 0.01$, $^{***}p < 0.001$, and $^{****}p < 0.0001$. **c** % mCerulean+ tumor cell: Day 1 MigR+Meso-CAR/Meso-CAR+Meso-CAR vs. FAP-CAR+Meso-CAR, $p = 0.001$ and 0.031, respectively. Day 7 Meso-CAR+Meso-CAR vs. FAP-CAR+Meso-CAR, $p = 0.004$. The $p$ values for remaining comparisons are all <0.001 or <0.0001. Source data are provided as a Source Data file.

T cells at the E:T ratios of 1:1, 5:1 and 10:1, in triplicate, in 96-well round-bottomed plates. After 18 hours, the culture supernatants were collected for IFN-γ analysis using an ELISA (Mouse IFN-γ Quantikine ELISA Kit, R&D System). The cytotoxic ability of FAP-CAR T cells was determined by detecting the remaining luciferase activity from the cell lysate using Luciferase Assay System (Promega) by measuring relative light unit (RLU). For checking the cytotoxicity of Meso-CAR T cells, 4662 mesothelin KO and parental 4662 PDAC cells were co-cultured with UTD, MigR or Meso-CAR T cells at the same E:T ratios as indicated above. Following an overnight co-incubation of T cells and target cells, supernatants were collected to quantify IFN-γ release by the same ELISA kit, and target cell viability was assessed using CellTiter 96 Aqueous Non-Radioactive Cell Proliferation Assay (MTS; Promega) by measuring the absorbance at 570 nm. In these killing assays, target cells alone were set as negative control to detect spontaneous death, and target cells lysed with water were set as positive control to get the maximal killing. The percentage of killing was calculated using the formula: % lysis = 100 × (spontaneous death - tested sample readout)/(spontaneous death - maximal killing).

**CAR T cell transmigration assay.** For transwell assays, the bottom wells were filled with 600 μl of T cell media with or without 100 ng/ml murine CXCL9 (Peprotech). $5 \times 10^5$ activated MigR control, Meso-CAR or FAP-CAR T cells were suspended in 100 μl T cell media and plated on the upper chamber of the transwell with 3 μm pore size filters (Millipore). After 4 hours of incubation at 37 °C, migrated T cells collected from the bottom wells were quantified using a cell counter.

**Tumorigenesis in PDAC transplant models.** For the syngeneic subcutaneous tumor models, 4662 or mCerulean⁺ 4662 ($3 \times 10^5$) PDAC cells were suspended into 100 mL PBS and inoculated s.c. into right flank of C57BL/6 mice, unless indicated otherwise in the Supplementary Fig. 3. For the human xenograft subcutaneous tumor model, human AsPC-1 or Capan-2 ($0.5 \times 10^6$) PDAC cells were co-injected with CCD-19 human lung fibroblasts ($1.5 \times 10^6$) suspended in 100 μL PBS and s.c. inoculated into right flank of NSG mice. Tumor size was measured three times per week using caliper. Tumor volume was calculated as width×width×length×0.5 and randomly assigned to the various treatment groups.

**CAR T therapies and combination therapies in PDAC transplant models.** For the monotherapy with CAR T cells, CD45.1 mouse MigR control, Meso-CAR, or FAP-CAR T cells were i.v. injected into tumor-bearing CD45.2 C57BL/6 mice at a dose of $5 \times 10^6$ CAR⁺ T cells/mouse when tumor volume reached 100–150 mm³; PBS was used as control ($n = 10$ per group), unless indicated otherwise in the Supplementary Fig. 3. Tumors were collected at 1- and 2-weeks post-T cell administration. For the survival analysis, mice were euthanized when tumor volume reached ~2000 mm³.

For the therapeutic combination of CAR T cells with anti-PD-1, CD45.1 mouse MigR control or FAP-CAR T cells were generated and i.v. injected into tumor-bearing CD45.2 C57BL/6 mice at a dose of $5 \times 10^6$ CAR⁺ T cells/mouse when the tumor volume reached 100–150 mm³. Isotype control or anti-PD-1 (BioXcell, 5 mg/kg twice a week) were i.p. injected into tumor-bearing mice 4 days post T cells administration ($n = 8$ per group). Tumor tissues were collected at day 17 post T cells administration for analysis. For the mice survival analysis, mice were euthanized when tumor volume reached ~2000 mm³.

For the dual doses of CAR T cell therapy, CD45.1 mouse MigR control, Meso-CAR or FAP-CAR T cells were generated and $5 \times 10^6$ CAR⁺ T cells/mouse were i.v. injected into tumor-bearing CD45.2 C57BL/6 mice or NSG mice when tumor volume reached 100–150 mm³ (syngeneic model) and 50–100 mm³ (xenograft models). Two weeks later, the mice treated with MigR control T cells received a second dose of $5 \times 10^6$ MigR T cells. Those treated with Meso-CAR T cells were treated with either MigR control or Meso-CAR T cells at a dose of $5 \times 10^6$ CAR⁺ T cells/mouse. The mice treated with FAP-CAR T cells were then given either MigR control, FAP-CAR, or Meso-CAR T cells at a dose of $5 \times 10^6$ CAR⁺ T cells/mouse. Tumor tissues were collected 31 days post T cells administration for analysis ($n = 5$ per group). For the modified survival analysis, mice were euthanized when tumor volume reached ~3200 mm³ in C57BL/6 mice or 800 mm³ in NSG mice.

**CAR T cells treatment of KPC mice bearing autochthonous PDAC tumors.** Fully backcrossed C57BL/6 KPC mice were monitored by ultrasound (Vevo 2100 Micro-Ultrasound, Visual Sonics, v1.4) and randomly assigned to the various treatment groups. $5 \times 10^6$ MigR control, CAR⁺ Meso-CAR or FAP-CAR T cells, were i.v. injected into mice with established pancreatic tumors of 50–100 mm³. For the monotherapy studies, tumor tissue was collected at 1- and 2-weeks post T cells administration. For the dual dose of CAR T cell therapies, $5 \times 10^6$ MigR control, CAR⁺ Meso-CAR or FAP-CAR T cells, were i.v. injected into KPC mice bearing established tumors of 50–100 mm³. Two weeks later, mice treated with either MigR control or Meso-CAR T cells were given a second dose of Meso-CAR T cells, while the mice treated with FAP-CAR T cells were given a second dose of FAP-CAR T cells or Meso-CAR T cells ($5 \times 10^6$ CAR⁺ T cells/mouse). For the survival analysis, mice were euthanized when tumor volume reached ~1000 mm³.

**Multiparametric flow cytometry of tumors, spleen, and peripheral blood cells**
Tumor tissue was dissected and digested with 2.5 mg/mL Collagenase type 2 (Worthington) plus 0.25 mg/mL DNase I (Roche) in RPMI-1640 for 30 min with intermittent shaking at 37 °C. Digestion mixture was passed through 70 μm followed by 40 μm cell strainers (FALCON) to prepare single cell suspension and washed with stain buffer (BD Biosciences). The spleens were mashed and passed through 40 μm cell strainer (FALCON). Red blood cells (RBC) were removed using RBC

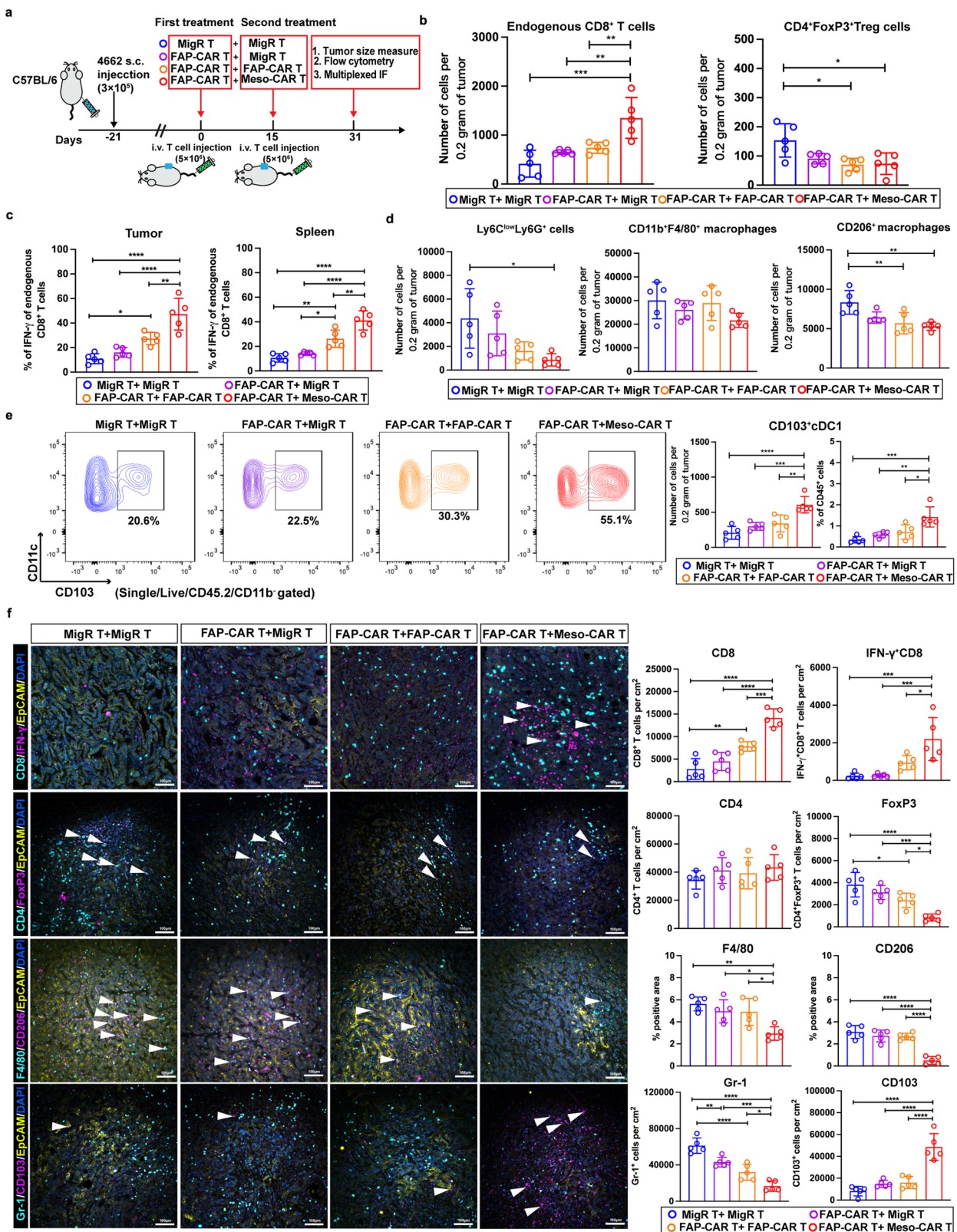

lysis buffer (BD Biosciences). Blood was collected through retro-orbital bleeding and RBC was removed using the same RBC lysis buffer. After washing with PBS, cells were collected.

Live/Dead Fixable Aqua Dead Cell Stain Kit (Invitrogen) was stained for 20 min on ice to exclude dead cells. Single cells were blocked with anti-mouse CD16/CD32 (Invitrogen) for 15 min followed by cell surface staining with antibodies diluted in stain buffer

(BD Biosciences) for 30 min on ice. For detecting intracellular cytokine, single cell suspensions were incubated for 5 hours with Cell Activation Cocktail (BioLegend) which is a pre-mixed cocktail with optimized concentration of PMA (phorbol 12-myristate-13-acetate), ionomycin, and protein transport inhibitor (Brefeldin A) at 37 °C. Intracellular staining for GzmB, IFN-γ, TNF-α, mesothelin, CD206 and TGF-β was performed with the use of Fixation/Permeabilization

**Fig. 8 | Sequential administration of FAP-CAR T cells with Meso-CAR T cells enhances systemic endogenous adaptive anti-tumor immunity in PDAC models. a** Treatment protocol. **b**–**d** Quantification of tumor-infiltrating endogenous CD8$^+$ T cells and CD4$^+$FoxP3$^+$ Treg cells IFN-γ expression in tumor-infiltrating and splenic endogenous CD8$^+$ T cells upon restimulation ex vivo Ly-6C$^{low}$Ly-6G$^+$ myeloid cells, CD11b$^+$F4/80$^+$ and CD206$^+$F4/80$^+$ macrophages by flow cytometric analysis. **e** Representative flow plots and quantification of flow cytometry analysis of CD103$^+$CD11c$^+$ cDC1 cells in tumors from the indicated combination treatment groups. **f** Representative images (left) and quantification (right) of multiplexed IF staining of sections of end-point tumors from each of the indicated treatment groups stained for CD8 (cyan) and IFN-γ (magenta), CD4 (cyan) and FoxP3 (magenta), F4/80 (cyan) and CD206 (magenta), Gr-1 (cyan) and CD103 (magenta) as indicated and in each case co-stained with EpCAM (yellow). Nuclei were stained with DAPI (blue). Scale bar, 100 μm. Data points are mean ± SD (n = 5 per group) and groups were compared using one-way ANOVA analysis with Dunnett's multiple comparison tests (**b**–**f**). $^*p < 0.05$, $^{**}p < 0.01$, $^{***}p < 0.001$, and $^{****}p < 0.0001$. **b** CD8:

FAP-CAR+MigR/FAP-CAR + FAP-CAR vs. FAP-CAR+Meso-CAR, $p = 0.003$ and 0.008, respectively. CD4$^+$FoxP3$^+$: all $p = 0.002$. **c** Tumor: MigR+MigR vs. FAP-CAR + FAP-CAR, $p = 0.015$; FAP-CAR + FAP-CAR vs. FAP-CAR+Meso-CAR, $p = 0.004$. Spleen: MigR+MigR vs. FAP-CAR + FAP-CAR, $p = 0.002$; FAP-CAR+MigR/FAP-CAR + FAP-CAR vs. FAP-CAR+Meso-CAR, $p = 0.016$ and 0.003, respectively. **d** Left: p = 0.018; right: MigR+MigR vs. FAP-CAR + FAP-CAR/FAP-CAR+Meso-CAR, $p = 0.008$ and 0.002, respectively. **e** Left: FAP-CAR + FAP-CAR vs. FAP-CAR+Meso-CAR, $p = 0.011$; right: FAP-CAR+MigR/FAP-CAR + FAP-CAR vs. FAP-CAR+Meso-CAR, $p = 0.003$ and 0.011, respectively. **f** CD8: MigR+MigR vs. FAP-CAR + FAP-CAR, $p = 0.004$. IFN-γ + CD8 +: FAP-CAR + FAP-CAR vs. FAP-CAR+Meso-CAR, $p = 0.022$. FoxP3: MigR+MigR vs. FAP-CAR + FAP-CAR, $p = 0.034$; FAP-CAR + FAP-CAR vs. FAP-CAR+Meso-CAR, $p = 0.019$. F4/80: MigR+MigR/FAP-CAR+MigR/FAP-CAR + FAP-CAR vs. FAP-CAR+Meso-CAR, $p = 0.001$, 0.014, and 0.018, respectively. Gr-1: MigR+MigR vs. FAP-CAR + FAP-CAR, $p = 0.006$; FAP-CAR + FAP-CAR vs. FAP-CAR+Meso-CAR, $p = 0.020$. The p values for remaining comparisons are all <0.001 or <0.0001. Source data are provided as a Source Data file.

---

Solution Kit (BD Biosciences), while Ki-67 and FoxP3 staining were performed with Foxp3/Transcription Factor staining buffer set (eBiosciences) according to the manufacturer's instructions. Antibodies used in flow analysis are described in Supplementary Table 1. Flow cytometry was performed on a Symphony A3 or LSR Fortessa flow cytometer (BD Biosciences, FACS Diva v6) and analyzed using FlowJo software (FlowJo LLC, USA).

### Multiplex immunofluorescence staining

Tumors removed from mice using forceps were fixed in Prefer fixation buffer (ANATECH) overnight at room temperature. Prefer-fixed paraffin-embedded (PFPE) sections were cut at 5 mm thickness, deparaffinized and rehydrated. Epitope retrieval was performed using citrate buffer (Sigma-Aldrich) using a pressure cooker (Bio SB) at low pressure (106-110 °C) for 5 min. PFPE sections were washed with PBS, incubated with blocking buffer (3%BSA with 0.3%Triton X-100 in PBS) for 1 hour at room temperature, and then stained with primary antibodies in blocking buffer at 4 °C overnight. Samples were stained with the following antibodies: FAP (1:250, Abcam), PDGFRα (1:200, R&D System), PDPN (1:200, R&D System), FITC-αSMA (1:500, Sigma), AF647-EpCAM (1:100, BioLegend), GFP (1:500, Abcam), CD3 (1:100, Abcam), CD8 (1:500, Abcam), CD4 (1:500, Abcam), FoxP3 (1:100, Abcam), F4/80 (1:100, Abcam), CD103 (1:100, Abcam), Ly6C + Ly6G (1:100, Abcam), CK19 (1:100, Abcam), Ki-67 (1:100, Abcam), CD31 (1:200, R&D System), IFN-γ (1:100, R&D System), TGF-β1 (1:100, Bioss), AF488 Pan-CK (1:100, eBiosciences), CD206 (1:100, R&D System), PD-1 (1:100, R&D System), biotin CHP (1:10, 3Helix), FN (1:300, Sigma-Aldrich), and cleaved caspase-3 (1:100, Cell Signaling Technology). Slides were then blocked again for 1 h and incubated with secondary antibodies in blocking buffer for 1 h, except for some that need to perform with tyramide signaling amplification (TSA) staining (Invitrogen). All primary, secondary antibodies and TSA kits used in multiplex IF staining are described in Supplementary Table 1. Slides were washed with PBS for once, incubated with DAPI for 5 min, then washed 3 times with PBS and mounted with ProLong™ Diamond Antifade Mountant (with DAPI, Invitrogen). The whole sections with multiplex IF staining were scanned by Nikon Ti-E inverted microscope (Nikon). NIS-Elements Advanced Research software (Nikon, version 4.50) was used to process the images and FIJI-ImageJ was used to analyze the images.

### Precision-cut tumor slice-based real-time two-photon microscopy

Tumor slices were prepared as described previously[30], with modifications. In brief, tumors were embedded in 5% low-gelling-temperature agarose (Sigma-Aldrich) prepared in PBS. Tumors were cut with a vibratome (Leica VT1200S vibratome) in a bath of ice-cold PBS. The thickness of the slices was 500 μm. Live tumor slices were stained with AF647-anti-mouse CD90.2 (BioLegend) at a concentration of 10 μg/mL for 15 minutes at 37 °C and were then transferred to 0.4-mm organotypic culture inserts (Millipore) in 35 mm Petri dishes containing 1 mL RPMI-1640 (without phenol red; ThermoFisher) before imaging.

Imaging fresh slices of mouse tumors was performed using Leica SP8-MP upright multiphoton microscope with Coherent Chameleon Vision II MP laser equipped with a 37 °C thermostatic chamber. Tumor slices were secured with a stainless-steel ring slice anchor (Warner Instruments) and perfused at a rate of 0.3 mL/min with a solution of RPMI (without phenol red), bubbled with 95% O$_2$ and 5% CO$_2$. Images were systematically acquired at 6 different regions within the tumor with a 20× (1.0 NA) water immersion lens and a Coherent Chameleon laser at 880 nm/25 mW. The following filters were used for fluorescence detection: CFP (483/32), GFP (535/30), AF647 (685/40) and tdTomato (610/75). For four-dimensional analysis of cell migration, a 70−90 μm z-stack at 5 μm step size was acquired for 2 hours, alternating between six fields every 30 seconds. Videos were made by compressing the z information into a single plane with the max intensity z projection of Imaris and LAS X software.

Image analysis was performed at PennVet Imaging Core (University of Pennsylvania). A 3D image analysis was performed on x, y, and z planes using Imaris 7.4 software. First, superficial planes from the top of the slice to 15 μm in depth were removed to exclude T cells located near the cut surface. Cellular motility parameters were then calculated using Imaris. Tracks >10% of the total recording time were included in the analysis. When a drift in the x, y dimension was noticed, it was corrected using the "Correct 3D Drift" plug-in in FIJI-ImageJ. CAR T cell number and motility were quantified in different tumor regions, including stroma-rich and tumor-nest regions. These regions were identified by visual inspection of immunofluorescence images. Stroma-rich region was defined by high CD90.2$^+$ area with a clear border between stromal cells and mCerulean$^+$ tumor cells. Tumor-rich region (nests) was defined by high Cerulean$^+$ area and stromal cells were interlaced with Cerulean$^+$ tumor cells. Fluorescence intensities were determined in regions of interest using FIJI-ImageJ. The number of T cells in defined regions was quantified using the Analyze Particles function of FIJI-ImageJ from fluorescent images that were first thresholded and then converted to binary images. Collagen measurement was performed using CT-FIRE software (version 2.0 beta) (https://loci.wisc.edu/software/ctfire)[58].

### Histopathology assessment

Animals were euthanized via carbon dioxide asphyxiation. Immediately after, blood was collected via cardiac puncture and kept in EDTA anticoagulant-coated tubes (Greiner Bio-One) until further processing. Hematological analysis was performed using IDEXX ProCyte Hematology Analyzer (IDEXX Laboratories, Westbrook, ME). Complete necropsy with macroscopic postmortem examination was performed

on all animals by the Comparative Pathology Core at the University of Pennsylvania School of Veterinary Medicine. Whole body weight was recorded during necropsy and the following tissues were collected and fixed in 10% neutral buffered formalin for histopathological examination: lungs, pancreas, spleen, sternum, and quadriceps femoris. Sternums were decalcified in EDTA tetrasodium salt solution before further processing using standard techniques as described before[59]. Formalin-fixed tissues samples were trimmed according to a standardized approach for rodent studies adapted from the RITA guidelines[60] and then routinely embedded in paraffin. Hematoxylin and eosin (H&E)-stained sections of the tissues were reviewed by a veterinary pathologist blinded to treatment conditions. The severity of lesions was graded on a semiquantitative scale from 0 (unremarkable or no change) to 4 (severe) in 1-point increments[60].

## Statistics and reproducibility

All experiments described here are representative of at least three independent experiments ($n \geq 5$ mice for each group unless specifically indicated). For in vitro experiments, cells, or tissues from each of these animals were processed (at least) in biological triplicates. All data here were shown as average±S.D. or average±S.E.M. Statistical analysis between two groups was conducted with a 2-tailed Student $t$ test. multiple comparisons were performed by using one-way ANOVA or two-way ANOVA analysis with Tukey's multiple-comparison. Tumor growth curve analysis was conducted with two-way ANOVA (mixed-model) with Tukey's multiple-comparison. Kaplan-Meier curves were used to analyze the survival data, and Cox regression was used to compute hazard ratio. No methods were used to determine whether the data met assumptions of the statistical approach. All statistical analysis was performed using GraphPad Prism 9 and the results of statistical analyses for each experiment are clearly indicated in the respective figures and in Figure Legends. $p$ values < 0.05 were considered significant.

## Reporting summary

Further information on research design is available in the Nature Portfolio Reporting Summary linked to this article.

## Data availability

The processed multiplex IF and 2-photon microscopy images and movies are available at Dryad [https://datadryad.org/stash/share/-WPXeg6_fw-CX-iSeRi3-BULdis7d0FbETaoNzMXx0Q]. The remaining data are available within the Article, Supplementary Information or Source Data file. Source data are provided with this paper.

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

## Acknowledgements

This work was supported by sponsored research agreements from TMUNITY Therapeutics to E.P and S.M.A, NIH/NCI P01 CA217805 to S.M.A. and E.P., and CRI Irvington fellowship (Grant#: CRI4168) from Cancer Research Institute to Z.X. The veterinary pathologists performing the histopathological analysis are part of the University of Pennsylvania Penn Vet Comparative Pathology Core Facility (RRID:SCR_022438) and are supported by the Abramson Cancer Center Support Grant (P30 CA016520). The PFPE sections were prepared by the Center for Molecular Studies in Digestive and Liver Diseases (P30DK050306) and the Molecular Pathology and Imaging Core at the University of Pennsylvania (RRID:SCR_022420).

## Author contributions

E.P. and Z.X. conceived the project and designed the experiments. Z.X., M.K., Y.L. and Li H. conducted the experiments and acquired the data. Z.X. generated CAR T cells, performed tumor killing assays and slice-based real-time two-photon microscopy. Z.W. and W.G. generated the mesothelin knockout line. Z.X., Z.L., Li H., N.P. and M.K. performed the mouse experiments. L.T., J.S. and E.P. designed and prepared the plasmids. Z.X., E.N-O., M.L. and C-A.A performed toxicity experiments and pathological analyses. Z.X. and Lili H. performed the statistical analysis of modified survival studies and interpreted the data. Z.X. and E.P. wrote the manuscript. E.P., C.H.J. and S.M.A provided guidance on experiments and edited the paper. All authors have read and approved the final manuscript.

## Competing interests

C.H.J., S.M.A. and E.P. are scientific founders and hold equity in Capstan Therapeutics. E.P. is on the scientific advisory boards of Parthenon Therapeutics and POINT Biopharma. S.M.A. is on the scientific advisory boards of Verismo and Bioardis. C.H.J. is a scientific founder and has equity in Tmunity Therapeutics/KITE Pharma and DeCART Therapeutics, reports grants from Tmunity Therapeutics, and is on the scientific advisory boards of BluesphereBio, Cabaletta, Carisma, Cellares, Celldex, ImmuneSensor, Poseida, Verismo, Viracta Therapeutics, WIRB Copernicus Group, and Ziopharm Oncology. S.M.A., E.P., L.T. and J.S. are inventors

(University of Pennsylvania) on a patent (10329355; Compositions and Methods for Targeting Stromal Cells for the Treatment of Cancer) and patent application for the 4G5 FAP CAR (Patent Applications 20210087294, Monoclonal Antibody Against Canine Fibroblast Activation Protein that Cross-Reacts with Mouse and Human Fibroblast Activation Protein [FAP] and 20210087295, Disrupting Tumor Tissues by Targeting Fibroblast Activation Protein [FAP]). S.M.A. and E.P. are inventors (University of Pennsylvania) on a patent application (US Provisional Patent Application 62/563,323 filed 26 September 2017, WIPO Patent Application PCT/US2018/052605: Targeting Cardiac Fibrosis with Engineered T Cells). In accordance with the University of Pennsylvania policies and procedures and our ethical obligations as researchers, C.H.J. is named on additional patents that describe the creation and therapeutic use of chimeric antigen receptors. These interests have been fully disclosed to the University of Pennsylvania, and approved plans are in place for managing any potential conflicts arising from licensing these patents. The remaining authors declare no competing interests.
