## [Peer Review File · Nature Communications]

Desmoplastic stroma restricts T cell extravasation and mediates immune exclusion and immunosuppression in solid tumorsEditorial Note: Parts of this Peer Review File have been redacted as indicated to maintain the confidentiality of unpublished data.

REVIEWER COMMENTS

Reviewer #1 (Remarks to the Author): with expertise in stroma/TME

Xiao and colleagues use subcutaneous pancreatic tumor grafts and the KPC genetically engineered PDAC model to study the effects of targeting FAP+ fibroblasts with FAP CAR T cells, in comparison with tumor cell-targeting CAR T (through mesothelin). They first show a better impact of FAP-CAR T on tumor regression compared to control or meso-CAR, associated with (1) a rapid decrease in FAP+ cell content and subsequent turnover/decrease in matrix molecules visualized by SHG, (2) a stronger CAR T cell entry into and motility/infiltration within the tumor lesion, (3) changes in the immune TME composition towards more endogenous CD8/NK cells and lower granulocyte-like LY6G+ cells. The authors then explore FAP-CAR T + anti-PD1 combination therapy, and a two-step therapeutic strategy combining stroma (FAP CAR) and tumor-cell (Meso CAR) targeting. The study provides a rationale to first target the tumor stroma making it favorable to T infiltration, for a subsequent higher efficacy of TAA CAR T cell therapy in desmoplastic solid tumors.

I find this manuscript very insightful and of high interest to the cancer immunology field now. Various therapeutic strategies aimed at targeting the stroma to overcome T cell exclusion have been developed in pre-clinical models, yet results from clinical trials have been mainly disappointing. Deciphering the mechanisms underlying stroma targeting successes and/or failures are needed.

The study here addresses some of these relevant and timely questions, and I would like to point out that the data is well analyzed and clearly described, with right controls and complementary quantification strategies. I only have few comments.

Main comment:

1) FAP can be expressed by healthy cells outside the tumor microenvironment. Can the authors analyze potential adverse effects (i.e. weight loss, anemia) and comment their work

and results in comparison with the study from D. Fearon (Roberts et al, JEM 2013) on depleting FAP+ cells.

Minor comments:

2) Fig 3b/c. Some discrepancies appear between the SHG images and quantifications. For example at D1 for the FAP CAR T condition in the stroma-rich regions the image shows almost no fibers (white) while the quantification shows 20-25% of SHG area/field. Likewise, in the tumor nest regions no fibers are observed on the images while 100-200 fibers/field are quantified. Could the authors explain these discrepancies?

3) Discussion, line 554. The ref 17 shows FAP+aSMA+ CAF association with T cell exclusion in human non small cell lung carcinoma, not PDAC.

4) Discussion, lines 581-585. The reference is missing.

Reviewer #2 (Remarks to the Author): with expertise in pancreatic cancer, therapy, stroma/TME

The manuscript by Xiao et al. demonstrates how FAP-targeted CAR T cells can complement the antitumor activity of both tumor antigen targeted CAR T cells (mesothelin) and immune checkpoint blockade with PD-1 targeted Ab in murine models of pancreatic cancer. They demonstrate compelling data that FAP-CAR T cells when administered alone or when combined with these other agents can actually deplete stromal cells, and physically interact with them, resulting in a modified tumor immune microenvironment. This leads to lasting changes marked by improved cytotoxic effector cells into these PDAC tumors. The data are quite impressive and show dramatic efficacy that is accompanied by immune modulatory action. Complementing these cell therapy studies are a series of experiments that also paired the FAP-CAR T cells together with PD-1 Ab blockade. These studies were done in mouse models that were refractory to anti-PD1 therapy. This data showed both efficacy as well as recruitment of endogenous host CD8 T cells into tumors.

The antitumor activity of dual FAP CAR T cells and Meso CAR T cells was particularly impressive, showing in some models regression of tumors (AsPC-1 model, Capan-2 model). They also document the effectiveness of sequential FAP targeted CAR T followed by Meso CAR T and demonstrate enhanced trafficking of these tumor antigen specific CAR T cells via this approach. Notably, changes in overall systemic phenotypes of CD8 T cells and type 1 DCs were also evident using this dual treatment approach.

In general the data contained within the manuscript is quite rigorous, with most conclusions drawn from results generated via at least two different methodologies such as flow cytometry and multiplex IF. In addition, it is a strength that in vivo efficacy studies used multiple models including KPC GEMM and they also used a short term tumor slice culture that removes tumors within 2 hr of harvest that produces superior data.

A few minor comments below can be addressed to further improve the manuscript:

1. Fig 1 should change the color scheme for FAP and Meso CAR T so differences can be better appreciated visually. The current color scheme has orange and red which are hard to discern from one another.
2. Typo, page 13, line 312, change 'unexpected' to 'unexpectedly'
3. Typo, page 24, line 589, change 'adaptively' to 'adoptively'
4. Overall the results should consider streamlining some sections of the discussion. In particular, sections that re-iterate the results section can be trimmed (for example, page 26 lines 625 – 635 are simply re-iterating results).
5. Can the authors speculate on the mechanism of systemic changes in immune response seen in CD8 and type I DCs? Do they think it is due to reduced tumor burden or cytokines or other factors derived from the CAR T cells that have been administered themselves?
6. Do the authors have any data on relative persistence of different Meso or FAP-targeted CAR T in vivo?

Reviewer #3 (Remarks to the Author): with expertise in CAR-T imaging

This paper deals with the mode of action of CAR T cells directed to fibroblast activation protein (FAP) in tumor pancreatic mouse models. The authors provide experimental

evidence that FAP-CAR T cells disrupt a suppressive stroma to increase the efficacy of PD-1 blockade and CAR T cells targeting a tumor-associated antigen. While prior studies including from the same group have started to define the antitumoral effects of FAP-CAR T cells in preclinical models, there is still a lot to learn about the mechanisms by which these CAR T cells function. Thus, the topic is significant and timely. They use many different approaches including multiplex-immunofluorescence and two-photon microscopy on fresh tumor slices to show that FAP-CAR T cells deplete FAP+ stromal cells leading to enhanced recruitment of CAR T cells and tumor growth control. Additional experiments demonstrate that this treatment remodels the tumor microenvironment from suppressive to stimulatory with more resident CD8 T and NK cells and fewer granulocyte-like myeloid cells. Finally, data are provided to suggest that FAP-CAR T cells enhance the efficacy of mesothelin-CAR T cells and anti-PD-1 therapy which are ineffective when used alone.

Overall, this is a well-performed study, and the data, for the most part, support the authors' conclusions. The conclusion that FAP-CAR T cells are sufficient to disrupt the suppressive TME of pancreatic tumors is novel. This is considered of major significance.

This manuscript could be improved in a few aspects, some of which are itemized below:

1) From their data obtained in vibratome slices, the authors conclude that FAP-CAR T cells overcome the immune exclusion. However, the quality of some of the images provided in this article makes it difficult to reach such a conclusion. The concept of immune exclusion is based on the compartmentalization of the tumor with tumor nests surrounded by a fibrous stroma and T cells sequestered in the stroma. The two PDAC models used in the current study - spontaneous and transplanted - show a good separation of both regions when visualized using IHC on paraffined sections (Figures 2G, H, 5G). However, this segregation is much less obvious in vibratome slices imaged with two-photon microscopy (Figure 3B and 7B). Moreover, data from supplementary figures 5B-D show the presence of SHG signals in both the stroma and tumor islets of vibratome slices which is not consistent with the IHC results reported here and in a previous study from the same group (Lo et al. Cancer Res 2015) showing matrix fibers exclusively in the stroma as it is the case in human pancreatic tumors.

- the authors should explain why the structure is different in the vibratome slices and the

IHC sections

- the authors should try improve the quality of their images obtained on vibratome slices (Figures 3B and 7B) that are sometimes hard to interpret.
- if problems persist, the calculation of the distribution of CAR T cells in the two compartments would be easier and more reliable to perform from the IHC images.
- to ensure increased recruitment of FAP-CAR T cells in tumor nests at different times, the ratio (number of T cells in tumor islets/number of T cells in the stroma) should be calculated.

2) The mechanism by which FAP-CAR T cells or meso-CAR T cells after FAP-CAR T cells are more numerous in tumors as compared to the other CAR T cells is not clear. Some data with CD31 immunostaining are provided to visualize CAR T cells in relation to vessels (supplementary Figure 5D). Again, the images are not of sufficient quality to conclude. In addition, the authors should reconcile their current data with their previous ones (Lo et al. 2015) showing that FAP-CAR T cells disrupt angiogenesis.

3) On several occasions, the authors state that FAP-CAR T cells in tumors are highly motile (lines 271, 281). From the data shown in Figures 3D and E, FAP-CAR T cells show a mean velocity of 0.2 $\mu\text{m}/\text{min}$ with very few cells reaching 1 $\mu\text{m}/\text{min}$. These velocity values are considered to be very low, indicating static cells. The average velocity of T cells in murine tumors usually reported ranges from 2 to 10 $\mu\text{m}/\text{min}$ depending on the tumors and the treatment (see for examples PMID: 22945631 and 25038199). Can the authors verify their values and ensure that they are correct?

4) In line 264, the authors state: "GFP+ MigR control and Meso-CAR T cells exhibited little, if any, stable interactions with stromal cells, matrix or tumor cells (Extended Data Movies 1-2) as evidenced by low levels of mean track speed". It is not possible to conclude this from the movies as stable cell-cell interactions are usually correlated with low cell motility.

Emmanuel Donnadieu

Reviewer #4 (Remarks to the Author): with expertise in CAR-T

Authors perform good investigation of fibroblast-activation protein (FAP)-directed CAR T cells and mesothelin CAR T cells using appropriate PDAC mouse models. Using a multitude of assays, they demonstrate significant observations. Authors are to be congratulated for extensive amount of work done for this project.

However, the manuscript weaknesses dominate than strengths -

1) Throughout the experiments, there is lack appropriate controls to support data interpretation. For example, by show of increased effector cytokines and survival following FAP CAR T cells, authors conclude FAP CAR T cells are more effective than meso CAR T cells. No cytotoxicity assay at similar E:T ratios is shown. The affinity of scFV / CAR and the antigen intensity of targets is not known to conclude one is superior to the other.

2) Attribution of all the benefits cannot simply to stromal destruction is questionable. On one hand, they describe the altered tumor microenvironment to effector following FAP CAR T-cell Rx, but then attribute second Rx with meso CAR T cells to stromal destruction and increased infiltration. Demonstration of higher number CAR T cells (accumulation) can be due to multiple factors of increased infiltration, proliferation or less exhaustion, cannot be simply attributed to infiltration.

3) The most important aspect of safety of FAP CAR T cells (on-target, off-tumor toxicity) for clinical translation is not addressed at all, not even in the discussion section. Without which the concept of targeting stroma by FAP to improve efficacy of antigen-targeted CAR T cells is not new as authors mention in the discussion section.

4) What is the cross reactivity of FAP CAR to mouse FAP? It is surprising that increased FAP CAR T cells in the PDAC tumor is seen within 5 days; tumor entry, antigen activation, cytokine secretion and lysis of stroma will take longer. Is it possible that the FAP CAR T cells were activated in the lung and thereby enter the tumor better (early)? One would think that if stromal destruction improved tumor infiltration, this to happen later than 5 days.

5) Did authors took into account the stromal content difference among tumors and the

outcomes (for example, the stroma in flank tumor versus orthotopic). Showing that the benefit is more pronounced in PDAC with more stroma (mimicking patient tumors) than no/less stroma (?flank tumor) may add support to the data interpretation.

6) Observation of less myeloid cells, increased extravasation are observations, cannot be directly attributed to stromal destruction without appropriate controls or focused experiments.

7) No experiments were conducted by administration of FAP and mesothelin CAR T cells together. Clinically, following cyclophosphamide lymphodepletion, administration of FAP CAR T cells results in their proliferation and consumption of cytokines leaving less room for mesothelin CAR T cells to proliferate. What will be the strategy to take this forward for translation?

8) Do authors have any observations with PDX models or human pancreatic tumors ex vivo?

9) Discussion section is lengthy with repetition of results than providing insights into limitations and potential translational strategies. If authors believe that the manuscript is providing insights into biological effects of stroma in shaping tumor immune microenvironment than translational CAR therapy, then that aspect should take priority.

Overall, good experiments / assays, and observations which do not support data interpretation or translational strategy. Without addressing safety of FAP CAR T cells, the observations noted are of interest, but not translational.

Response to reviewers' comments:

Response to Reviewer #1:

We are grateful to the reviewer for the kind summary of our study and that they found the study “very insightful and of high interest to the cancer immunology field”, that it addresses “relevant and timely questions,” and that the data are “well analyzed and clearly described....”. We also appreciate the suggestions made by this reviewer. We believe that incorporating their suggestions as described below in our point-by-point response, has improved the manuscript.

Main comment:

1) FAP can be expressed by healthy cells outside the tumor microenvironment. Can the authors analyze potential adverse effects (i.e. weight loss, anemia) and comment their work and results in comparison with the study from D. Fearon (Roberts et al, JEM 2013) on depleting FAP+ cells.

We agree that it is important to consider the potential adverse off-tumor on-target effects of depleting FAP⁺ cells and we have addressed this issue at some length in prior publications (see references below). For the reviewers' benefit, we summarize this issue in detail here and describe new data and discussion of this issue that have been incorporated into the revised manuscript under consideration:

Multiple groups, including our own, have demonstrated using a variety of approaches (e.g. CAR T cells, antibody drug conjugates, etc.) that transiently depleting a majority of FAP^{high} cells is sufficient to inhibit tumor growth without any apparent toxicity (Wang LS. et al. *Cancer Immunol Res.* 2014. PMID: 24778279; Lee IK, et al. *Clin Cancer Res.* 2022. PMID: 35972732; Kakarla S. et al. *Mol Ther.* 2013. PMID: 23732988; Liu Y. et al. *J Transl Med.* 2023. PMID: 37046312). For example, in each of our prior studies and each of the experiments described herein, we monitored body weight and never observed weight loss in tumor-bearing mice that received a single or two doses of FAP-CAR T cells compared to PBS, MigR-control vector, or Meso-CAR T cell treated mice. Body weight measurements from two representative experiments are shown in Expanded Data Figs. 2c and 3a-b of the revised manuscript. In some studies, we also performed extensive pathological analyses of multiple organs (New Extended Data Fig. 3b), blood biochemistry studies and WBC of tumor-bearing mice 1 week following a single dose of T cells and in a separate cohort, 1 week post a second dose of T cells (new Extended Data Fig. 3c). The data shown in new Extended Data Fig. 3 are based on the blinded observations, and in the case of extended Figs. 3b and 3c, the blinded observations of a certified veterinary pathologist who has been added as a co-author. Findings included minimal bone marrow hypoplasia in just 2 out of 5 mice that received a single dose, and no bone marrow changes in any of the mice one week after two doses of FAP-CAR T cells. Mild to moderate splenic hyperplasia was observed in 60% of mice that received a single dose and 100% of mice given two doses of FAP-CAR T cells. Mild to moderate mononuclear infiltrate of quadriceps muscle, pancreas (with a single mouse exhibiting more severe infiltrate) and lung were observed at 1 week post a single injection of FAP-CAR T cells, but these effects appeared to be transient, as they for the most part resolved by 1 week post a second dose of FAP-CAR T cells. Although changes in ALP, RBC, HGB, and HTC were noted, they were similar between control MigR and FAP-CAR T cell treated mice and therefore related to adoptive cell

transfer and not related to the activity of FAP-CAR T cells per se, as was the case for the changes in WBC as well (new Extended Data Fig. 3c). Overall, minimal long-term toxicity was observed. These data are referred to in the results section of the revised manuscript on page 7, line 159-160 and page 8, line 173-191.

Consistent with our current and prior studies, Kakarla and colleagues showed anti-tumor efficacy without toxicity of an independently engineered anti-FAP-CAR human T cells comprising the scFV from the M036 mAb that targets both human and mouse FAP in an immune-deficient mouse model of human lung cancer (Kakarla S. et al. *Mol Ther.* 2013 PMID: 23732988). Similarly, a recent study using yet another independently engineered anti-FAP CAR based on antibody 8E3, also showed efficacy without any evidence of toxicity (Liu Y. et al. *J Transl Med.* 2023. PMID: 37046312).

However, as expected, this approach has a therapeutic window. Exceeding this therapeutic window by complete ablation of rare normal FAP^{low} populations, as well as fibrogenic FAP^{high} cells, in particular cancer associated fibroblasts (CAFs), can result in defined toxicities in pre-clinical mouse models. With regard to studies published by Fearon's group, it is important to note that their results were obtained using a mouse model in which FAP⁺ cell depletion was induced by administration of diphtheria toxin to Bac transgenic mice expressing DTR under the control of a transgenic FAP promoter rather than the endogenous FAP promoter. In their initial report based on this approach, Fearon's group demonstrated inhibition of tumor growth in response to conditional ablation of FAP⁺ cell without notable toxicity (Kraman M, et al. *Science.* 2010. PMID: 21051638.). Only upon more detailed analysis of weight loss, bone marrow and skeletal muscle in the subsequent paper by Robert's et al (Roberts EW, et al. *J Exp Med.* 2013. PMID: 23712428) referred to by the reviewer, did they observe evidence of adverse effects of FAP⁺ cell ablation. It is likely that the difference in toxicity in their model is, at least in part, due to the fact that their genetic ablation approach resulted in the complete elimination of FAP⁺ cells including both cells expressing high levels of FAP (eg. CAFs), as well as the few populations of normal cells (eg. bone marrow cells, alpha-glucagon producing pancreatic islet cells, and mesenchymal stem cells) that typically express significantly lower levels of FAP (2-5 fold in mouse and 5-10 fold in human; data not shown). In contrast, our CAR T cell approach which may in fact be translatable to clinical settings, selectively kills cells expressing moderate to high levels of FAP while sparing FAP^{low} cells. Consistent with this interpretation, we could recapitulate to some extent the bone marrow acellularity, weight loss and pancreatic perivascular inflammatory infiltrate, by exceeding the therapeutic window by repeated dosing with hyper-functional FAP-CAR T cells (Wang LS. et al. *Cancer Immunol Res.* 2014. PMID: 24778279). Importantly, based on the data in the current study, we expect that FAP-CAR T cells will mainly be exploited as a dual therapy in which their synergy with other immune-based or chemotherapies will require fewer and/or lower doses and transient functionality (for example by using mRNA CAR T cells) of FAP-CAR T cells, than would be required to achieve inhibition of tumor growth as a monotherapy, thus also minimizing risk of toxicity. Other possible but not mutually exclusive contributing factors include epitope specificity, differing affinities of distinct CARs and CAR constructs. Thus, we contend that there are multiple ways to achieve selectivity for FAP^{high} cells and to tune the efficacy of FAP-CAR T cells, and/or limit CAR expression, to keep within the therapeutic window to safely inhibit tumor growth.

A second important consideration is that some of the toxicities observed in mouse models may or may not be conserved in human. We have now suggested this in the revised discussion. Although beyond the scope of this manuscript, and therefore not discussed in detail in the revised manuscript, but for the reviewers' benefit, we summarize the findings of our extensive analysis of FAP expression across virtually all human tissues at the mRNA and protein levels. First, we have noted that levels of expression and distribution of FAP in mouse bone marrow is not conserved in human bone marrow where expression is lower and restricted to a very small population of hematopoietic cells and not evident on the bone lining cells that prominently express FAP in mouse bone marrow. Second, whereas expression on mouse CAFs was found to be 2-5 fold less than seen on any of the normal FAP^{low} populations, this differential expression was exaggerated in human where CAFs expressed 5-10 fold higher levels of FAP than normal FAP^{low} cells.

Finally, it should be noted that a clinical trial of intrapleurally administered FAP-CAR T cells has been conducted in mesothelioma patients with minimal toxicity and a suggestion of efficacy (Hiltbrunner S, et al. *Ann Oncol.* 2021. PMID: 33098996). A trial of intravenously administered FAP-CAR T cells is underway in China.

Minor comments:

2) Fig 3b/c. Some discrepancies appear between the SHG images and quantifications. For example, at D1 for the FAP CAR T condition in the stroma-rich regions the image shows almost no fibers (white) while the quantification shows 20-25% of SHG area/field. Likewise, in the tumor nest regions, no fibers are observed on the images while 100-200 fibers/field are quantified. Could the authors explain these discrepancies?

We appreciate the reviewer's perception of an apparent discrepancy. Fig. 3b represents a merged image of the multiplexed imaging that obscures the SHG signal. The quantification, however, was based on analysis of the SHG channel using CT-FIRE software for the readout of fiber number and ImageJ software for the readout of SHG areas. The **non-merged images used for quantification** are depicted in now Extended Data Fig. 7b-d (related to Fig. 3b/c). In the non-merged images provided, both thick and aligned fibrillar collagen as well as the mesh-like collagen visualized by SHG can easily be detected in these sections.

3) Discussion, line 554. The ref 17 shows FAP+ α SMA+ CAF association with T cell exclusion in human non-small cell lung carcinoma, not PDAC.

We thank the reviewer for pointing out this error. The correct reference for this statement that was intended to be cited is Öhlund D, et al. *J Exp Med.* PMID: 28232471, and is cited as such in the relevant section of the introduction. However, as the result of shortening the discussion at the reviewers' request, the pertinent statement is no longer included in the discussion.

4) Discussion, lines 581-585. The reference is missing.

We thank the reviewer for their careful reading. As the result of shortening the discussion at the reviewers' request, the pertinent statement is no longer included in the discussion.

Response to **Reviewer #2**:

In general, the data contained within the manuscript is quite rigorous, with most conclusions drawn from results generated via at least two different methodologies such as flow cytometry and multiplex IF. In addition, it is a strength that in vivo efficacy studies used multiple models including KPC GEMM and they also used a short-term tumor slice culture that removes tumors within 2 hr of harvest that produces superior data.

A few minor comments below can be addressed to further improve the manuscript:

The reviewer's positive comments regarding the rigor including the use of multiple models and orthogonal approaches as well as optimized assays are highly appreciated. We also thank the reviewer for making suggestions that further help improve our manuscript.

1. Fig 1 should change the color scheme for FAP and Meso CAR T so differences can be better appreciated visually. The current color scheme has orange and red which are hard to discern from one another.

The reviewer's point has been well taken. We reformatted the color of FAP- and Meso-CAR schemes to provide a better visualization for the readers.

2. Typo, page 13, line 312, change 'unexpected' to 'unexpectedly'

3. Typo, page 24, line 589, change 'adaptively' to 'adoptively'

We thank the reviewer for their careful reading of the manuscript and regret the noted typos that have been corrected in our revised manuscript.

4. (The reviewer suggests we) consider streamlining some sections of the discussion. In particular, sections that re-iterate the results section can be trimmed (for example, page 26 lines 625 – 635 are simply re-iterating results).

As the reviewer suggested, we have trimmed these sections of the discussion in our revised manuscript.

5. Can the authors speculate on the mechanism of systemic changes in immune response seen in CD8 and type I DCs? Do they think it is due to reduced tumor burden or cytokines or other factors derived from the CAR T cells that have been administered themselves?

We fully agree with the potential contributing factors raised by the reviewer and although beyond the scope of the current manuscript, we plan to explore these potential mechanisms in future studies.

Nonetheless, we have briefly speculated on this issue by adding the following comments to the revised discussion (see page 25, line 593-597): “Of interest, we also observed systemic changes in immune response in CD8 and type I DCs, which might result either from reduced tumor burden and or from the induced release of cytokines or other factors by the activated CAR T cells. Future studies will determine the contributions of these potential mechanisms.”

6. Do the authors have any data on relative persistence of different Meso or FAP-targeted CAR T in vivo?

We thank the reviewer for raising this question. We did, in fact, study this in a head-to-head comparison of each CAR T (see figure). After i.v. injection of each CAR T, we performed flow cytometry on single cell suspensions prepared from whole tumors at various time-points. As depicted in new Extended Data Fig. 6 of the revised manuscript (and shown here), we found greater numbers of FAP-CAR T cells accumulated intra-tumorally compared to Meso-CAR T cells at each time point assessed. Moreover, we found that FAP-CAR T cells also persisted somewhat longer.

Response to **Reviewer #3**:

We thank the reviewer for the overall positive comments and especially their appreciation of our use of multiple models, orthogonal approaches to analysis and the novelty and significance of our finding regarding impact of FAP-CAR T cells on the ECM. Please find our point-by-point response to their valuable comments below.

1. From their data obtained in vibratome slices, the authors conclude that FAP-CAR T cells overcome the immune exclusion. However, the quality of some of the images provided in this article makes it difficult to reach such a conclusion. The concept of immune exclusion is based on the compartmentalization of the tumor with tumor nests surrounded by a fibrous stroma and T cells sequestered in the stroma. The two PDAC models used in the current study - spontaneous and transplanted - show a good separation of both regions when visualized using IHC on paraffined sections (Figures 2G, H, 5G). However, this segregation is much less obvious in vibratome slices imaged with two-photon microscopy (Figure 3B and 7B).

- the authors should explain why the structure is different in the vibratome slices and the IHC sections

- the authors should try to improve the quality of their images obtained on vibratome slices (Figures 3B and 7B) that are sometimes hard to interpret.

We agree that the segregation is somewhat more difficult to appreciate in the images of the thick vibratome slices. This is due, in part, to loss of resolution through the processing of these tissues for

imaging as explained below, and to some extent a reflection of an arguable disadvantage of 3D- vs 2D imaging. Specifically, for the IHC sections, we performed antibody-based staining on Prefer-fixed paraffin-embedded thin-sections. Due to the fluorescence bleaching caused by fixation and tissue processing and the fluorescent signal from the mCerulean-expressing tumor cells, these samples were then stained post-processing with either pan-CK or anti-EpCAM cell surface reactive antibodies. Multiplexed IF staining to simultaneously detect tumor cells and stromal cells (anti-FAP) results in high contrast imaging of the tumor border and thus clearly delineating the tumor nests from surrounding fibrous stroma in thin sections. In contrast, the vibratome sections were not subjected to fixation, and in this assay, we relied on the endogenous more diffuse cytoplasmic mCerulean tag expressed by the tumor cells, in combination with anti-CD90.2 staining of stromal cells to distinguish these two compartments. These differences in methods, in combination with the impact of 3D-imaging of thick sections by 2-Photon microscopy (that relies on merged multiple Z-sections) that focus on a single plane relative to epifluorescent imaging of thin sections, further blurred the visualization of the border between these two compartments, as pointed out by the reviewer. It is important to note however, that even with this caveat, the essential point that treatment with FAP-CAR T cells disrupted the interface and immune exclusion compared to controls, was highly evident by both methods. We continue to work to optimize the 3D-imaging approach.

Moreover, data from supplementary figures 5B-D show the presence of SHG signals in both the stroma and tumor islets of vibratome slices which is not consistent with the IHC results reported here and in a previous study from the same group (Lo et al. Cancer Res 2015) showing matrix fibers exclusively in the stroma as it is the case in human pancreatic tumors.

This point pertains to the issue also raised by Reviewer 1 comment 2. Briefly, as described in our response to reviewer 1, Fig. 3b represents a merged image of multiplexed imaging. Merging the images from all the channels, unfortunately obscured the SHG signal and as a result is not obvious in Fig 3b. We refer the reviewer to Extended Data Figs. 7b-d (related to Fig. 3b/c) in which we provide the image from the SHG channel only that was used for the quantification of the signal using CT-FIRE software for fiber number and ImageJ software to determine SHG area of such **non-merged images**. In the non-merged images, both thick and aligned fibrillar collagen, as well as the mesh-like collagen visualized by SHG, can easily be detected in these sections. The data analysis in Lo et al. cited by the reviewer was **based solely on antibody-based IHC and not SHG imaging** which we have found to be more sensitive for detecting, in particular, thin collagen fibers that we find predominate within tumor nests, compared to the wide range of collagen network, thin fibers and thick collagen bundles found surrounding the tumor nests.

-if problems persist, the calculation of the distribution of CAR T cells in the two compartments would be easier and more reliable to perform from the IHC images

.- to ensure increased recruitment of FAP-CAR T cells in tumor nests at different times, the ratio (number of T cells in tumor islets/number of T cells in the stroma) should be calculated

Indeed, we collected serial tumor slices for IHC which were spatially close to the live samples that we used from matched tumors for the 2-photon microscopy assay. With the use of multiplexed IHC staining, we

further validated and confirmed our findings. We also calculated the distribution of CAR T cells in the two compartments and the ratio (number of T cells in tumor islets/number of T cells in the stroma) based on the IHC staining and have added these additional data as Extended Data Fig. 7e to validate the analyses of thick slices by 2-photon live image that is presented in Figure 3.

2) The mechanism by which FAP-CAR T cells or meso-CAR T cells after FAP-CAR T cells are more numerous in tumors as compared to the other CAR T cells is not clear. Some data with CD31 immunostaining are provided to visualize CAR T cells in relation to vessels (supplementary Figure 5D). Again, the images are not of sufficient quality to conclude. In addition, the authors should reconcile their current data with their previous ones (Lo et al. 2015) showing that FAP-CAR T cells disrupt angiogenesis.

The greater accumulation of FAP-CAR T cells is likely multifactorial. The reviewer correctly points out that we previously reported a decrease in angiogenic vessels. The data in Fig. 6 of the current study, showing a reduction in CD31 staining following FAP-CAR T cells treatment, confirm our earlier report by Lo et al. However, this is not necessarily inconsistent with an increase in accumulation of T cells post-treatment with FAP-CAR T cells since it is now well-established that there is not a direct relationship between vascular density and immune/inflammatory cell infiltrate, as the latter is also critically dependent on vascular integrity and function, as well as physical parameters such as intra-tumoral pressure, ECM composition and the physical properties of the ECM such as stiffness etc. Although beyond the scope of this paper and therefore only included here for the reviewer's benefit, we are currently further investigating the relationship between the impact of FAP-CAR T cells on the vasculature and T cell infiltration into tumors. In any case, we do clearly demonstrate in this study that FAP-CAR T cells have a greater capacity to infiltrate tumor and therefore have a greater opportunity to interact with their cognate antigen resulting in their further survival and proliferation/accumulation.

With regard to the strength of the data now in Extended Data Fig. 7f (originally in Extended Data Fig. 5d) depicting the spatial relationship of fluorescently tagged CAR T cells to blood vessels detected by antibody staining of CD31 of thick sections and visualized by live 2-photon microscopy, we refer the reviewer to the responses to comment 2 from Reviewer #1 and comment 1 above from this reviewer. In addition, we have now also performed IHC in paired sections from the same tumors. Quantification of extravasated T cells in these high resolution 2D images validated our original conclusion based on the 3D images in Extended Data Fig. 7f. These new complementary data are included in the revised manuscript as Extended Data Fig. 7g.

3) On several occasions, the authors state that FAP-CAR T cells in tumors are highly motile (lines 271, 281). From the data shown in Figures 3D and E, FAP-CAR T cells show a mean velocity of 0.2 $\mu\text{m}/\text{min}$ with very few cells reaching 1 $\mu\text{m}/\text{min}$. These velocity values are considered to be very low, indicating static cells. The average velocity of T cells in murine tumors usually reported ranges from 2 to 10 $\mu\text{m}/\text{min}$ depending on the tumors and the treatment (see for examples PMID: 22945631 and 25038199). Can the authors verify their values and ensure that they are correct?

We appreciate the reviewer astutely picking up on this. We made an error in labeling of the Y axis in Figs. 3d and 3e. The decimal was misplaced so that the axis read 0-1.5 when they were meant to be one log greater dynamic range, i.e. 0-15 $\mu\text{m}/\text{min}$. After carefully checking all raw data and readouts from Imaris software, we have now corrected these labels in revised Fig. 3.

4) In line 264, the authors state: “GFP+ MigR control and Meso-CAR T cells exhibited little, if any, stable interactions with stromal cells, matrix or tumor cells (Extended Data Movies 1-2) as evidenced by low levels of mean track speed”. It is not possible to conclude this from the movies as stable cell-cell interactions are usually correlated with low cell motility.

The relative propensity of GFP⁺ MigR and Meso-CAR T cells compared to FAP-CAR T cells to form stable interactions is of interest. However, we agree with the reviewer that the extended data movies 1-2 are not sufficient to reach the conclusion that their propensity to do so is reduced and, in fact agree that the low mean track speed might imply the opposite to be true. The most compelling evidence we have for this conclusion is instead that, although we could readily detect stable interactions of Meso-CAR T with tumor cell targets (Fig. 6) in mice pre-treated with FAP-CAR T cells, we never observed through careful screening of multiple movies, any stable cell-cell interactions involving Meso-CAR T cells in mice treated with Meso-CAR T cells alone. This is in contrast to the data in now in Extended Data Fig. 7h (originally in Extended Data Fig. 5f) in the revised manuscript, in which we describe the readily detectable stable interactions of FAP-CAR T cells with target cells in mice treated with FAP-CAR T cells alone. We have readdressed this argument in this section (page 13, line 293-294) of the results.

With regard to the reduced mean track speed of the GFP+ MigR and Meso-CAR T cells compared to the FAP-CAR T cells, we speculate that this is due to the fact that only the FAP-CAR T cells engage their cognate antigen resulting in their activation as evidenced by the blast morphology, prominent europods and increased motility, features not observed for the MigR or Meso-CAR T cells.

Response to **Reviewer #4**:

Authors perform good investigation of fibroblast-activation protein (FAP)-directed CAR T cells and mesothelin CAR T cells using appropriate PDAC mouse models. Using a multitude of assays, they demonstrate significant observations. Authors are to be congratulated for extensive amount of work done for this project.

We are grateful for the reviewer’s positive comments.

However, the manuscript weaknesses dominate than strengths –

We acknowledge there remain some interesting questions raised by this study, but hope that we have addressed many of them in our revision and in the response to the reviewers’ comments, while others we

believe are beyond the scope of this manuscript, but will be addressed in our ongoing and future research for publications.

Throughout the experiments, there is lack appropriate controls to support data interpretation. For example, by show of increased effector cytokines and survival following FAP CAR T cells, authors conclude FAP CAR T cells are more effective than meso CAR T cells. No cytotoxicity assay at similar E:T ratios is shown. The affinity of scFV / CAR and the antigen intensity of targets is not known to conclude one is superior to the other.

Although the reviewer is correct that the efficacy of two CAR T cells targeting the same antigen on the same target cells (i.e. two different FAP-CAR T cells targeting cancer-associated fibroblasts [CAFs]) can be compared, there are important caveats to comparing the efficacy of two CARs that target different antigens on different target cells (i.e. FAP-CAR T killing of CAFs vs Meso-CART killing of tumor cells). Each target cell will have a different density of target antigen, as well as its own intrinsic resistance to T cell-mediated killing. With these caveats in mind, we compared the relative intrinsic functionality of the FAP-CAR T cells and Meso-CAR T cells *in vitro* and the results of these studies are shown in detail in Fig. 1 and Extended Data Fig. 1. This includes data from a cytotoxicity assay where the cytolytic capacity of each of the CAR T cells against their respective antigen expressing target cells (Fig. 1c) and interferon-gamma release (Fig. 1d) was determined at multiple E:T cell ratios (Fig. 1c), as suggested by the reviewer. These data establish that the two CAR T cells indeed have similar antigen-specific activity and activation *in vitro*, thus allowing us to attribute their differential efficacy *in vivo* to extrinsic factors introduced by the tumor microenvironment and/or the host ecosystem having differential impact on the two CAR T cells and thus, support our conclusions that the FAP-CAR T cells are “better” than Meso-CARTs with respect to inhibition of tumor growth. In any case, the most impactful aspect of the paper is in fact that treatment with FAP-CAR T cells improved the efficacy of subsequent Meso-CAR T injection, a conclusion which is not in any way dependent on their comparable intrinsic activity. In our studies in which we treated with both CAR T cells, we had all the proper controls (and a large number of them) clearly showing that the anti-tumor efficacy of Meso-CAR T cells was much better after pretreatment with FAP-CAR T cells.

2) Attribution of all the benefits cannot simply to stromal destruction is questionable. On one hand, they describe the altered tumor microenvironment to effector following FAP CAR T-cell Rx, but then attribute second Rx with meso CAR T cells to stromal destruction and increased infiltration. Demonstration of higher number CAR T cells (accumulation) can be due to multiple factors of increased infiltration, proliferation or less exhaustion, cannot be simply attributed to infiltration.

We completely agree with the reviewer that the impact of FAP-CAR T cells is not likely due only to the depletion of the stromal matrix but also critically dependent on the direct effects of depleting FAP⁺ stroma cells and the indirect effects of depleting the FAP⁺ stromal cells and matrix on the behavior of multiple other tumor components including immune cells, endothelial cells, malignant epithelial cells and other subpopulations of stromal cells such as the classic myofibroblasts that were also depleted at later timepoints. Defining all the mechanisms involved in the complex cross-talk between FAP⁺ stromal cells including the role matrix, and the multitude of other components of the tumor microenvironment is

clearly beyond the scope of a single manuscript. For example, as indicated in our response to comment number 6 below from this reviewer, there may be other mechanisms in place, such as changes in secretion of chemokines, growth factors, or cytokines caused by the removal of the fibroblasts, all of which we will investigate going forward. Nonetheless, we did identify at least three of the mechanisms by which depletion of FAP⁺ cells exerted its anti-tumor activity, and more importantly how treatment with FAP-CAR T cells rendered tumors vulnerable to the tumoricidal activity of Meso-CART cells, as stated in the in abstract and now also stated more clearly in the results and discussion sections of the revised manuscript. Specifically, we present evidence to support our conclusions that FAP-CAR T cell efficacy reflects their capacity to 1) overcome stroma-dependent restriction of T cell extravasation and/or perivascular invasion, 2) reverse immune exclusion, and 3) relieve T cell suppression resulting in alterations in the immune landscape i.e., reducing myeloid cell accumulation and increasing endogenous CD8⁺ T cell and NK cell infiltration and function. To clarify what might be a semantic issue, in referring to stroma in these statements, we are referring collectively to the FAP⁺ stromal cell and the extracellular matrix, both of which are depleted in the FAP-CAR T treated tumors; we have clarified our use of this term in the revised manuscript.

We also agree with the reviewer that just seeing a higher number of CAR T cells within tumors could be the result of many factors including trafficking, retention, or proliferation. However, the real novelty and strength of this study is that we visualized the tumors in real time using 2-photon microscopy so we could differentiate these contributing factors. Our data clearly show that Meso-CAR T cells traffic more efficiently after FAP-CAR T cell-mediated stromal disruption and this is not due to increased proliferation or even increased retention.

3) The most important aspect of safety of FAP CAR T cells (on-target, off-tumor toxicity) for clinical translation is not addressed at all, not even in the discussion section. Without which the concept of targeting stroma by FAP to improve efficacy of antigen-targeted CAR T cells is not new as authors mention in the discussion section.

We thank the reviewer for raising this critical issue. We discuss this issue in considerable detail above and refer the reviewer to our response to Reviewer 1, Comment 1 that also raised this issue. We have also added additional data (new Expanded Data Fig. 3) and discussion (page 26) regarding toxicity to the revised manuscript.

4) What is the cross reactivity of FAP CAR to mouse FAP? It is surprising that increased FAP CAR T cells in the PDAC tumor is seen within 5 days; tumor entry, antigen activation, cytokine secretion and lysis of stroma will take longer. Is it possible that the FAP CAR T cells were activated in the lung and thereby enter the tumor better (early)? One would think that if stromal destruction improved tumor infiltration, this to happen later than 5 days.

We apologize for not making the species specificity clear. The 4G5 anti-FAP antibody and the FAP-CAR used in this study derived from the 4G5 scFv sequence targets cross-reacts with mouse and human FAP. We have now added this piece of information into the second sentence in the first section of “Results”.

We agree with the reviewer that the rapidity of the effects we saw was surprising. In our experience using human CAR T cells, we usually don't see anti-tumor effects for a number of weeks. Moreover, the impact of tumor antigen-targeted mouse CAR T cells that have been the focus of prior CAR T cells studies in mouse models of solid tumors have shown very modest and sometimes more delayed effects. Importantly, in addition to analyzing the behavior of TAA specific MESO-CAR T cells using flow cytometry and static IF staining of fixed sections, the direct visualization we performed in this study clearly demonstrate their poor tumor infiltration thus delaying and reducing the maximal number of MESO-CAR T cells that accumulate in the tumors consistent with the later response expected based on prior studies. In stark contrast, consistent with our experience with mouse CAR T cells in syngeneic mice, and in particular with stroma cell targeted CAR T cells, as well as the new flow cytometry and IF data presented in the current paper, the live imaging of the FAP-CAR T cells using 2-photon microscopy directly demonstrate for the first time that FAP-CAR T cells infiltrate tumors much more rapidly and as a result the kinetics of their impact on tumor stromal cells and tumor growth are accelerated. It is interesting to note that this kinetics also support our conclusion (Lo A, et al. *Cancer Res.* 2015. PMID: 25979873) that depletion of FAP⁺ cells inhibits tumor growth through both immune-independent mechanisms that can happen more rapidly, as well as immune-dependent mechanisms which would not be expected to take effect until at least 1 week post-treatment. Frankly, we also were initially surprised at how quickly the matrix was disrupted following treatment with FAP-CAR T cells, but the data were robust, consistent, and reproducible across many experiments with multiple controls. Indeed, we now think that this data regarding the dynamics of a pathologic matrix is one of the most fundamental and surprising findings of our study that provide important and novel insights into the dynamics of tumor-associated desmoplastic matrix. Moreover, we have found similar rapid dissolution of matrix in a cardiac fibrosis model following treatment with FAP-CAR T cells (Rurik JG, et al. *Science.* 2022. PMID: 34990237). We have hypotheses as to the mechanisms that underly this exciting finding that are currently being tested in the laboratory and that we expect will be the focus of a future manuscript.

5) Did authors take into account the stromal content difference among tumors and the outcomes (for example, the stroma in flank tumor versus orthotopic). Showing that the benefit is more pronounced in PDAC with more stroma (mimicking patient tumors) than no/less stroma (flank tumor) may add support to the data interpretation.

We present data obtained with tumors generated from the KPC-derived 4662 cell line transplanted subcutaneously into syngeneic mice and autochthonous tumors that developed spontaneously in the pancreases of KPC mice. Although there is tumor to tumor variability in the amount of stroma in both of these models, in general they are both quite stroma-rich with the spontaneous tumors tending to have a somewhat higher stromal content. Interestingly, we did note that the efficacy of FAP-CAR T cells also tends to be greater in the spontaneous pancreatic tumor model. It is also worth noting that in prior studies of other mouse tumor models including lung tumors and mesothelial tumors, that had more modest to moderate stroma compared to these pancreatic models, we also saw significant but less robust inhibition of tumor growth (Wang LS, et al. *Cancer Immunol Res.* 2014. PMID: 24778279). This correlation also held up in human xenograft models where the inhibition of PDAC tumor growth shown in this study was more robust than we previously observed in A549 lung tumors. Furthermore, consistent with this concept, we

found FAP-CAR T cells had no impact on the growth of human mesothelial I45 tumors that elicit little if any stromal response. Although collectively our cumulative data are consistent with the reviewer's suggestion, since differences other than the stromal content may also impact the efficacy of FAP-CAR T cells, we feel more work will be required before we can definitively conclude that there is a direct relationship between stromal content and FAP-CAR T cell efficacy.

6) Observation of less myeloid cells, increased extravasation are observations, cannot be directly attributed to stromal destruction without appropriate controls or focused experiments.

We agree with the reviewer that there may be other mechanisms in place, such as changes in secretion of chemokines, growth factors, or cytokines caused by the removal of the fibroblasts. To avoid overstatement, we have now rephrased the corresponding sentences throughout the revised manuscript.

7) No experiments were conducted by administration of FAP and mesothelin CAR T cells together. Clinically, following cyclophosphamide lymphodepletion, administration of FAP CAR T cells results in their proliferation and consumption of cytokines leaving less room for mesothelin CAR T cells to proliferate. What will be the strategy to take this forward for translation?

We thank the reviewer for emphasizing the translational potential of our proposed combination strategy. We agree that from a translational standpoint, it would be easier to administer both the FAP and CAR T cells together. Accordingly, we have begun studies to test this idea. The preliminary data below, show our first study, which looks promising. We believe this line of inquiry is beyond the scope of this paper and plan to develop it more fully for a second manuscript.

[FIGURE REDACTED]

Another potential way forward could be to generate transient FAP-CAR T cells *in vivo* by delivering modified messenger RNA (mRNA) in T cell-targeted lipid nanoparticles (LNPs) and then follow-up with a tumor targeted CAR. As proof of principal, we recently showed that treatment with modified mRNA-targeted FAP-CAR mRNA LNPs reduced fibrosis and restored cardiac function after injury (Rurik JG, et al. *Science*. PMID: 34990237).

8) Do authors have any observations with PDX models or human pancreatic tumors *ex vivo*?

In this current work, we have used two different human PDAC cell lines (Capan-2 and AsPC-1) in the context of a xenograft approach. We consistently observed complete tumor regression in the tumors treated with the combination of FAP-CAR followed by Meso-CAR T cells in these models (see Fig. 6). We also are trying to obtain human pancreas cancer samples for future studies in order to conduct studies on precision cut tumor slices (PCTS), similar to those performed on mouse tumors in this study. However, initially we performed studies using the standard method of cutting tumor slices and then treating them with a therapeutic reagent (in our case CAR T cells) and monitoring the fate of the CAR T cells and the impact of the treatment on tumors over time (on the order of days) *ex vivo*. Interestingly, we noted a very robust wound response due to slicing the tumors in the control samples. So, although we observed a

robust impact of our CAR T cells using this approach, our analysis suggested that the effects were as much on the wound response as to the tumors themselves and thus made it difficult to interpret the precise impact relative to the impact they would have on an “uninjured” tumor *in vivo*. In the mouse system, we were able to modify the approach by treating with the CAR T cells *in vivo* and then harvesting tumors at different times after treatment and slicing and imaging/analyzing the impact immediately (within hours) after tumor harvest before the onset of a wound response. Of course, this adaptation to this otherwise elegant standard approach to studying human PCTS is not feasible and therefore we deprioritized using this approach to study human tumors. We also deprioritized studies of PDXs due to evidence that this approach although very valuable for studying the behavior of tumor cells, does not faithfully recapitulate the stromal compartment (Liu Y, et al. *Signal Transduct Target Ther.* 2023. PMID: 37045827). So, although we acknowledge the limitations of the xenograft model, due to these technical considerations, it is the best we can do readily at this juncture.

9) Discussion section is lengthy with repetition of results than providing insights into limitations and potential translational strategies. If authors believe that the manuscript is providing insights into biological effects of stroma in shaping tumor immune microenvironment than translational CAR therapy, then that aspect should take priority.

We have extensively trimmed the discussion by ~25% based on the reviewer’s suggestion.

Overall, good experiments / assays, and observations which do not support data interpretation or translational strategy. Without addressing safety of FAP CAR T cells, the observations noted are of interest, but not translational.

We have addressed the safety related issues above and have added new toxicity data to the revised manuscript as described above. The fact that FAP-CAR T cells are already in clinical trials shows the translational potential of this approach.

REVIEWERS' COMMENTS

Reviewer #1 (Remarks to the Author):

I thank the authors for the added data and comments, which fully address my questions and concerns. Their findings are for sure exciting.

Reviewer #3 (Remarks to the Author):

I thank the authors for having performed significant additional work and providing an adequate response to all my comments.

Could you, please, check the legend in supplementary figure 7G? The legend states that the FAP-CAR T cells (green) located within 10 mm (0-10 mm) or beyond (10-25 mm) the closest CD31+ blood vessel (gray) in the entire area of tumor section were counted. mm is probably not the correct measurement.

Reviewer #4 (Remarks to the Author):

[none]

Response to reviewers' comments:

Response to Reviewer #1:

I thank the authors for the added data and comments, which fully address my questions and concerns. Their findings are for sure exciting.

We thank the reviewer for the positive comments on our revisions.

Response to Reviewer #3:

I thank the authors for having performed significant additional work and providing an adequate response to all my comments.

Could you, please, check the legend in supplementary figure 7G? The legend states that the FAP-CAR T cells (green) located within 10 mm (0-10 mm) or beyond (10-25 mm) the closest CD31+ blood vessel (gray) in the entire area of tumor section were counted. mm is probably not the correct measurement.

We have corrected the typographical error pointed out by the reviewer by changing the unit of measurement from "mm" to "µm" in our revised manuscript.